# The archaeal protein SepF is essential for cell division in *Haloferax volcanii*

Phillip Nußbaum[1], Maren Gerstner[1], Marie Dingethal[1], Celine Erb[1] & Sonja-Verena Albers [1✉]

In most bacteria, cell division depends on the tubulin homolog FtsZ and other proteins, such as SepF, that form a complex termed the divisome. Cell division also depends on FtsZ in many archaea, but other components of the divisome are unknown. Here, we demonstrate that a SepF homolog plays important roles in cell division in *Haloferax volcanii*, a halophilic archaeon that is known to have two FtsZ homologs with slightly different functions (FtsZ1 and FtsZ2). SepF co-localizes with both FtsZ1 and FtsZ2 at midcell. Attempts to generate a *sepF* deletion mutant were unsuccessful, suggesting an essential role. Indeed, SepF depletion leads to severe cell division defects and formation of large cells. Overexpression of FtsZ1-GFP or FtsZ2-GFP in SepF-depleted cells results in formation of filamentous cells with a high number of FtsZ1 rings, while the number of FtsZ2 rings is not affected. Pull-down assays support that SepF interacts with FtsZ2 but not with FtsZ1, although SepF appears delocalized in the absence of FtsZ1. Archaeal SepF homologs lack a glycine residue known to be important for polymerization and function in bacteria, and purified *H. volcanii* SepF forms dimers, suggesting that polymerization might not be important for the function of archaeal SepF.

[1] Molecular Biology of Archaea, Institute of Biology II, Faculty of Biology, University of Freiburg, Freiburg, Germany. ✉email: sonja.albers@biologie.uni-freiburg.de

Most archaeal and bacterial cells divide by binary fission, resulting in two daughter cells of equal size and DNA content. Cell division in most bacteria starts with the polymerization of the conserved tubulin homolog FtsZ into a ring-like structure at midcell. Polymerization of FtsZ proteins into single-stranded protofilaments is initiated after GTP binding[1]. The so-called Z-ring is associated with the cell membrane, providing a scaffold for downstream cell division proteins, many of them involved in septal peptidoglycan (PG) synthesis[2–5]. However, FtsZ lacks a membrane-targeting domain and is therefore dependent on proteins that tether the FtsZ polymers to the membrane. In *Escherichia coli* FtsA and ZipA are responsible for the membrane association of FtsZ filaments. ZipA contains an N-terminal trans-membrane domain and is unique to Gammaproteobacteria[6]. Like ZipA, FtsA binds to the conserved C-terminal peptide of FtsZ[7–9]. FtsA, a protein that forms actin-like protofilaments in the presence of ATP, binds to the membrane via a C-terminal amphipathic helix[9–11]. *Bacillus subtilis* lacks ZipA but has, besides FtsA[12] and EzrA[13], another FtsZ membrane anchor protein called SepF[14]. SepF is highly conserved in Firmicutes, Actinobacteria, and Cyanobacteria and acts as an alternative membrane anchor for FtsZ with an N-terminal amphipathic helix[15,16]. Yet, SepF binds to the conserved C-terminal peptide of FtsZ promoting the assembly and bundling of FtsZ polymers[17–19]. A knock-out of *sepF* in *B. subtilis* resulted in deformed division septa which led to the proposal that SepF is required for a late step in cell division[16]. This suggests that, besides anchoring FtsZ filaments to the membrane, SepF also has a regulatory role in cell division. Indeed it was discovered that the overexpression of SepF in *B. subtilis* leads to complete delocalization of late cell division proteins[20] and SepF overproduction in *Mycobacterium smegmatis* led to filamentation of the cells[21].

Electron microscopy of purified SepF from *B. subtilis* showed polymerization into large ring-like structures with an average diameter of about 50 nm in vitro[22]. The observation of liposomes bound exclusively to the inside of SepF rings led to the assumption that in vivo the protein does not form rings but arcs, perpendicularly associated to the nascent septum[15]. The ability of SepF to form rings in vitro is conserved in bacteria. A recent investigation of SepF homologs from different bacteria indicated that the SepF ring diameter correlates with the septum width[23], supporting the model that SepF forms a clamp on top of the leading edge of the growing septum.

The crystal structure of SepF showed a dimer of two SepF monomers that contain a compact α/β-sandwich of two α-helices stacked against a five-stranded β-sheet. Dimers are formed by the interaction of the β-sheets and polymerization is acquired by the interaction of the α-helices of adjacent dimers[15]. The highly conserved residue G109 of *B. subtilis* is important for the interaction between dimers since a mutation in this residue leads to SepF dimers that do not polymerize[15]. However, a recent study of SepF from *Corynebacterium glutamicum* showed that dimerization of SepF monomers is also possible by the interaction of the lateral two α-helices forming a four-helix bundle. Moreover, it was observed that in this conformation two hydrophobic pockets are formed that enable interaction with FtsZ[19].

In archaea, cell division systems are much less uniform than in bacteria. Three very different cell division mechanisms have been identified in archaea: (i) a mechanism based on actin homologs, (ii) a mechanism based on homologs of the eukaryotic ESCRT-III system, and (iii) a mechanism involving FtsZ[24]. Studies on archaeal cell division have so far mainly focused on the crenarchaeal Cdv-system that is based on homologs of the ESCRT-III system also found in eukaryotes[25]. Additionally, there are some studies on FtsZ-based euryarchaeal cell division, especially from Haloarchaea[26–32]. Haloarchaea often contain multiple proteins belonging to the FtsZ/tubulin superfamily[33]. A study in *Haloferax volcanii* showed that only two of the eight proteins from this superfamily are true FtsZ homologs. Both FtsZ homologs from *H. volcanii*, FtsZ1 and FtsZ2, were shown to be important for cell division and fulfill slightly different functions in the process[34]. The remaining proteins form a distinct phylogenetic group named CetZ. Members of the CetZ group are not involved in cell division but in controlling the cell shape[35].

Interestingly, a bioinformatical approach identified putative archaeal SepF homologs in all FtsZ containing archaea, most of them belonging to the euryarchaeal superphylum[24]. Putative archaeal SepF homologs from *Archaeglobus fulgidus* and *Pyrococcus furiosus* were used previously for crystallization studies[15]. However, no functional studies on archaeal SepF homologs are available and it is currently unknown if these homologs are actually involved in cell division.

In this study, we used *H. volcanii* to characterize a putative archaeal SepF homolog. *H. volcanii* is perfectly suited for the investigation of euryarchaeal cell biology due to the availability of an advanced genetic system[36] and functional fluorescent proteins[35], together with its easy cultivation conditions. Fluorescent microscopy of GFP-tagged SepF showed that it is localized at the cell center together with FtsZ1 and FtsZ2. Additionally, the gene is essential and characterization of its function was only possible upon depletion. SepF-depleted *H. volcanii* cells strongly increased in size over time and showed a cragged cell surface. On the other side, the presence of plasmids in the SepF depletion strain leads to filamentation of the cells upon SepF depletion. In contrast, additional production of SepF had no effect on the growth and shape of *H. volcanii* cells. Moreover, localization of FtsZ1 and FtsZ2 in SepF-depleted cells might point to another so far unknown FtsZ1 membrane anchor. Indeed, an interaction of SepF with FtsZ2 but not with FtsZ1 was shown. In contrast to bacterial SepF, the archaeal SepF does not form polymers. In summary, we provide evidence that SepF is a key player in cell division in archaea using the FtsZ system.

## Results

**Localization of SepF in *H. volcanii*.** The first analysis indicating that FtsZ containing archaea might have SepF homologs was conducted by Makarova et al. in 2010[24]. In this study, archaeal clusters of orthologous genes (arCOGs) were identified with the same phyletic pattern as the arCOG containing the archaeal FtsZ homologs. arCOG 02263 was identified as a cluster that seemed to contain archaeal SepF orthologs. This cluster is located on the main chromosome and contains the *hvo_0392* gene from *H. volcanii*. Interestingly, *hvo_0392* is co-transcribed with the downstream gene *hvo_0393*[37,38], which encodes a putative DNA repair ATPase. Like *hvo_0393*, the other adjacent genes of *hvo_0392* encode proteins that are most likely not involved in cell division (Supplementary Fig. 1a).

The protein encoded by *hvo_0392* is composed of 118 amino acids with a molecular weight of 12.6 kDA (Supplementary Fig. 1b). Comparison of the sequence of this protein with SepF sequences from selected bacteria and archaea showed a weak sequence similarity between the archaeal proteins and the SepF proteins from bacteria. Some of the residues of SepF, which were described to be important for interaction with FtsZ[15,19], were conserved in both phyla (Supplementary Fig. 1c). However, G109 of *B. subtilis* SepF which is important for oligomerization[15] and is conserved in other bacterial SepFs, is not present in archaeal SepF homologs (Supplementary Fig. 1c).

To establish whether Hvo_0392 is involved in cell division, the gene was expressed with a C-terminal GFP-tag under the control of a tryptophan inducible promotor located on a plasmid

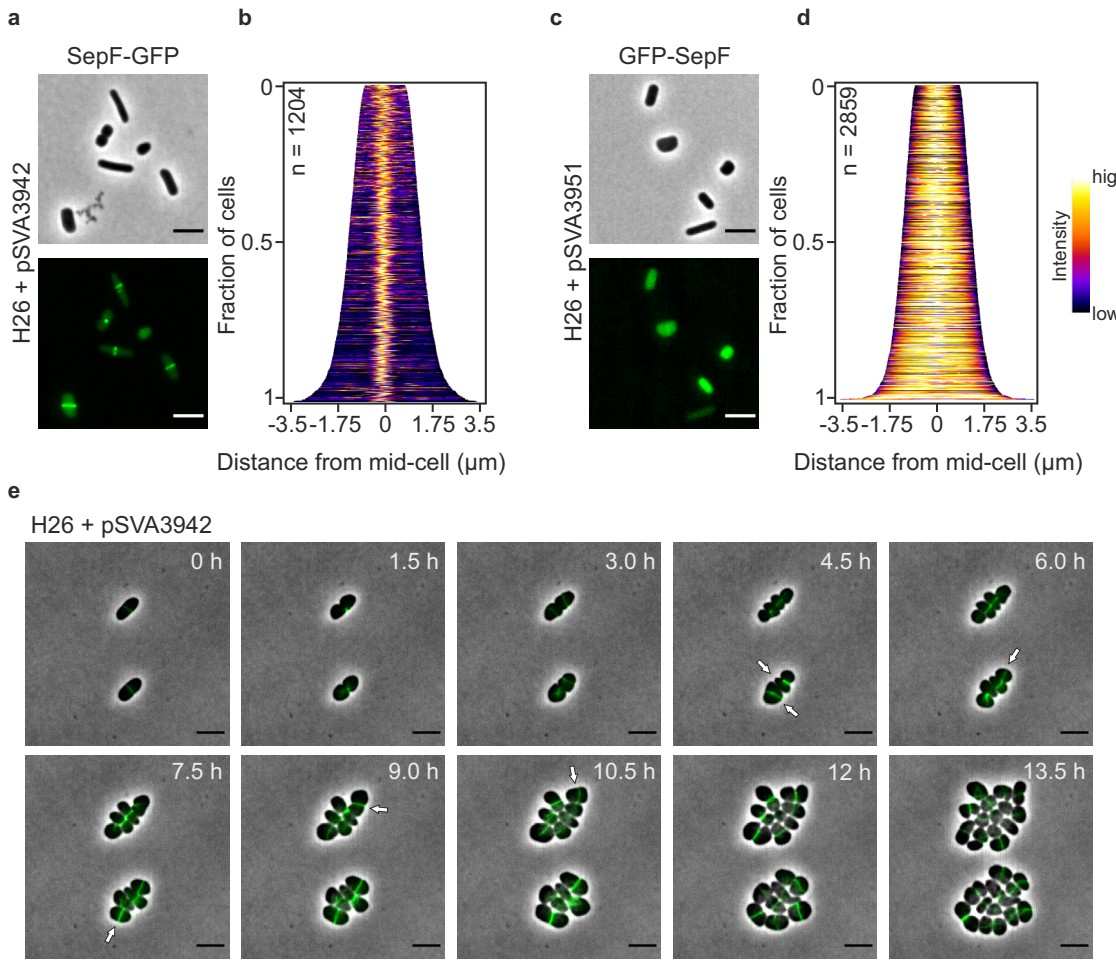

**Fig. 1 Localization and dynamic of SepF from _H. volcanii_. a** Fluorescence microscopy of SepF-GFP expressing cells during early exponential growth. **b** Demographic analysis of the GFP signal in the imaged cells shows localization of SepF in the cell center independent of the cell length in all cells. **c** Fluorescence microscopy of GFP-SepF expressing cells during early exponential growth. **d** Demographic analysis of the GFP signal in the imaged cells shows the diffuse signal of SepF throughout the cells independent of the cell length. Demographs were based on the GFP signals of cells imaged in three independent experiments. **e** Time-lapse microscopy of SepF-GFP expressing cells. Cells were grown on an agarose nutrition pad in a thermo-microscope set to 45 °C and imaged every 30 min. The selected pictures show constricting SepF rings during cell division. The new cell division plane in the daughter cells often occurs perpendicular to the old one (indicated by white arrows). The experiment was repeated three times independently with similar results. Scale bars: 4 μm.

(pSVA3942) in laboratory _H. volcanii_ wild type strain H26. Fluorescence microscopy showed localization of the protein in a ring-like structure at midcell (Fig. 1a), at a similar location as observed for FtsZ1 and FtsZ2 in _H. volcanii_[34,35]. At the resolution used, it was not possible to determine whether the rings were continuous or assembled from patches (Supplementary Fig. 2). Localization of the tagged protein at the cell division plane in all observed cells (Fig. 1b) strongly indicated that the protein encoded by the gene _hvo_0392_ is involved in cell division and might have a similar function as bacterial SepF. Therefore, Hvo_0392 is referred to as SepF in this manuscript. SepF was localized at midcell in all imaged _H. volcanii_ cells independent of the cell length, indicating that SepF is directly localized to the cell center after cell division. This was observed in both short cells (2 μm in length), that had just gone through cell division and elongated cells (6 μm in length), that were close to division (Fig. 1b). In contrast, expression of SepF with an N-terminal GFP-tag showed diffuse localization throughout the cytoplasm (Fig. 1c). From bacteria, it is known that SepF contains an N-terminal membrane targeting side (MTS)[15,23]. Most likely the diffuse localization of GFP-SepF was caused by the GFP-tag

blocking the MTS of the archaeal SepF from interaction with the cell membrane (Fig. 1d).

To obtain insight into the dynamic behavior of the _H. volcanii_ SepF homolog, time-lapse microscopy was performed. _H. volcanii_ cells expressing SepF-GFP were monitored for 13 h in a temperature-controlled microscope. During that time SepF rings were observed to constrict in the middle of the cells during cell division. Immediately after septum closure, new SepF rings appeared at the future cell division plane in the daughter cells, often perpendicular to the old division plane (Fig. 1e, Supplementary Movie 1). To further investigate the influence of the N-terminal region of the archaeal SepF on the localization of the protein, the putative MTS sequence and SepF without that sequence were fused to GFP. Interestingly, the short peptide (MGIMSKILGGGG) forming the MTS of SepF was sufficient to efficiently localize GFP to the cell center in ring-like structures, as observed for the full-length SepF above (Fig. 2a, b). Moreover, SepF without its MTS was also shown to form ring-like structures at midcell possibly by interaction with endogenous SepF (Fig. 2c). However, in only 20% of the observed cells (n = 3329) the construct was detected and showed localization at midcell. In

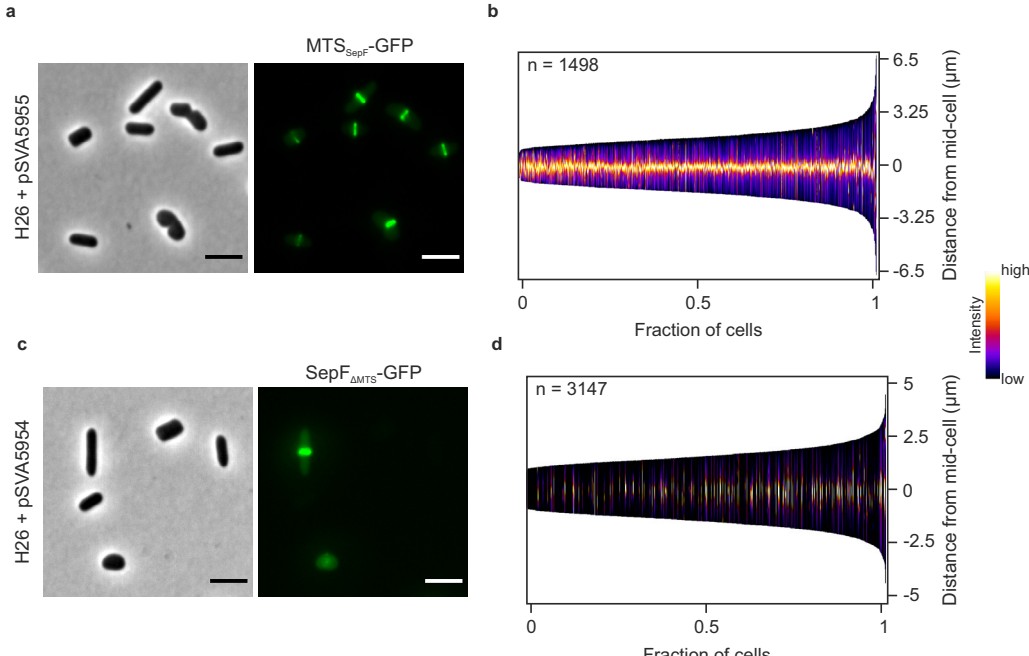

**Fig. 2 Localization of only the membrane-targeting site of SepF and SepF without its membrane binding site. a** Fluorescence microscopy of MTS$_{SepF}$-GFP in *H. volcanii* cells during early exponential growth. **b** Demographic analysis of the GFP signal in the imaged cells shows localization of the GFP-tagged membrane-targeting site of SepF at midcell, independent of the cell length in all observed cells. **c** Fluorescence microscopy of SepF-GFP with the MTS of SepF deleted. **d** The construct was only expressed in 20% of the cells. However, in these cells SepF without MTS was localized at midcell regardless of the cell length. Both demographs summarize the results of three independent experiments per construct. Scale bars: 4 μm.

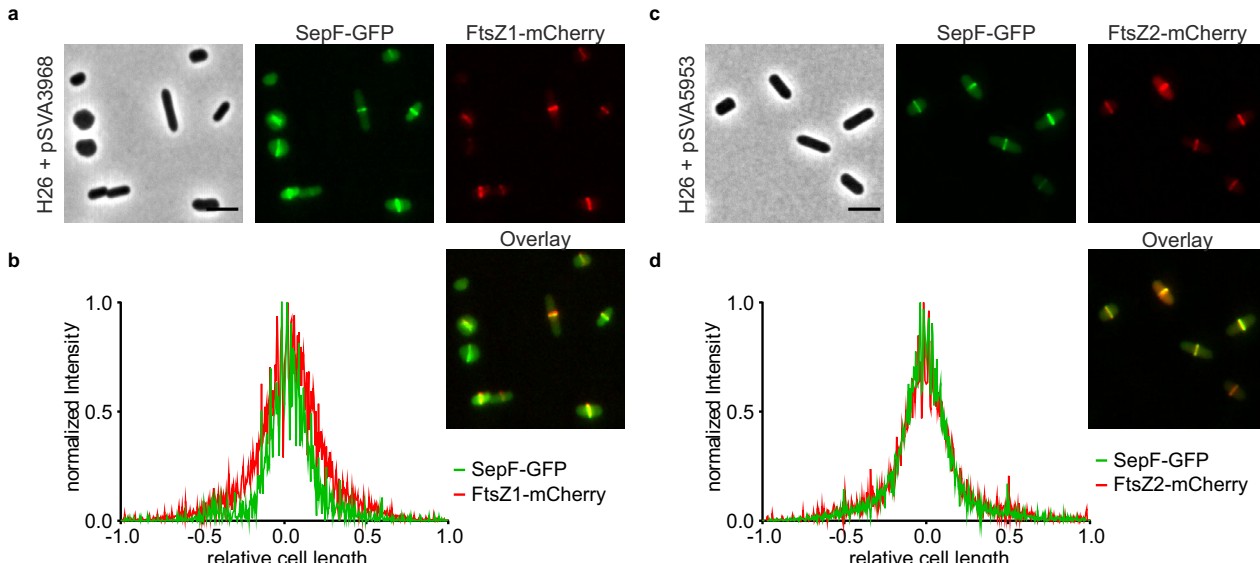

**Fig. 3 Co-localization of SepF with either FtsZ1 or FtsZ2. a** Fluorescence microscopy of *H. volcanii* cells expressing SepF-GFP and FtsZ1-mCherry in early exponential phase. **b** Intensity profile of the normalized fluorescence signal from the SepF-GFP (green) and the FtsZ1-mCherry (red) construct relative to the cell length. **c** Fluorescence microscopy of *H. volcanii* cell expressing SepF-GFP and FtsZ2-mCherry in early exponential phase. **d** Intensity profile of the normalized fluorescence signal from the SepF-GFP (green) and the FtsZ2-mCherry (red) construct relative to the cell length. The experiment was repeated at least three times per construct with >1000 cells analyzed. Scale bars: 4 μm.

most of the cells, SepF-GFP without its MTS was not detected (Fig. 2d).

**Co-localization of SepF with FtsZ1 and FtsZ2.** Since SepF-GFP was observed at midcell during cell division, it was tested whether it colocalizes with the tubulin homologs FtsZ1 and FtsZ2 at midcell. Therefore, the respective genes were cloned in a double expression vector, expressing SepF with a C-terminal GFP-tag and FtsZ1 or FtsZ2 with a C-terminal mCherry-tag (Fig. 3a, c). Both FtsZ1 (Fig. 3b) and FtsZ2 (Fig. 3d), localized together with SepF at midcell. However, while the signals of SepF and FtsZ2 strongly overlapped, the FtsZ1 signal was slightly broader than the SepF signal. These results show that SepF localizes like FtsZ1 and FtsZ2 at midcell during cell division, indicating that SepF might be a part of the divisome of *H. volcanii*.

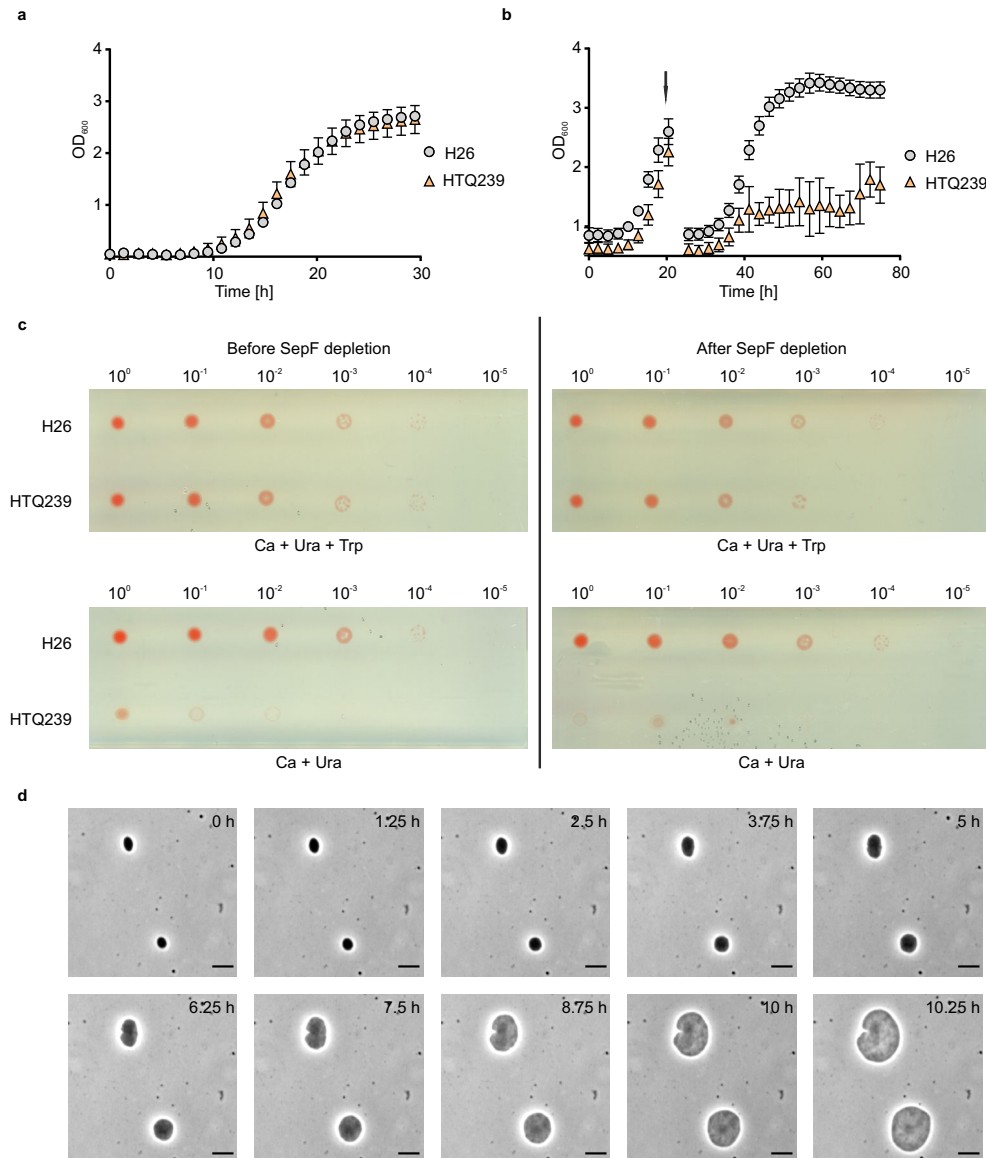

**Fig. 4 Growth of H26 and HTQ239 in the presence and absence of tryptophan. a** Growth curve of H26 (gray) and SepF depletion strain HTQ239 (orange) grown in CAB-medium supplemented with tryptophan and uracil. **b** Growth curve of H26 (gray) and HTQ239 (orange) after dilution in CAB-medium without tryptophan (indicated by the arrow). After the transfer, HTQ239 stopped growing while H26 grew unaffected. The data for these growth curves were acquired from four independent cultures per strain. Error bars indicate the standard deviation. **c** Spot survival assay of H26 and HTQ239. In the left panel, cells were cultured under non-SepF depleting conditions before they were spotted on plates with or without tryptophan. On the right panel, cells were grown under depleting conditions before they were spotted on plates with or without tryptophan. Under both culturing conditions of the precultures, strain HTQ239 showed the same viability of H26 when grown on plates supplemented with tryptophan. In contrast, on plates without tryptophan the viability of strain HTQ239 dropped compared to H26. In total three technical replicates with three biological replicates per strain and pre-culture condition were performed. **d** Time-lapse microscopy of HTQ239 cells in the absence of tryptophan: Cells were grown on an agarose nutrition pad (without tryptophan) in a thermo-microscope set to 45 °C and imaged every 15 min. During imaging the cells strongly increased in size while they were not able to execute cell division. Exemplarily cells every 1.25 h after depletion are shown. The experiment was repeated three times independently with similar results. Scale bars: 4 μm.

**SepF is essential in _H. volcanii_.** To obtain insight into the cellular function of archaeal SepF several attempts to delete _sepF_ were made. However, we were not able to generate a knock-out mutant of _sepF_ suggesting an essential function for SepF. To test this hypothesis, the native promotor of the gene was exchanged by a tryptophan inducible promotor resulting in the strain HTQ239. The HTQ239 strain showed normal growth when grown in selective medium complemented with 0.25 mM tryptophan (Fig. 4a). After dilution into selective medium without tryptophan the culture stopped growing whereas the control

strain H26 grew unaffected (Fig. 4b). To assess the viability of the SepF depleted cells spot dilution assays were performed (Fig. 4c). HTQ239 cells that were grown in liquid under non-depleting conditions showed no difference to H26 cells when spotted on plates supplemented with tryptophan (Fig. 4c, left upper panel). In contrast, on plates without tryptophan the depletion strain failed to grow (Fig. 4c, left lower panel). As expected cells that were already grown under SepF depletion conditions in liquid also did not grow on plates lacking tryptophan. However, SepF depleted HTQ239 that was spotted back on plates supplemented

with tryptophan grew like the control strain H26 indicating that the absence of SepF leads to a halt in cell division, but the cells remain viable (Fig. 4c, right upper panel).

To further study the effects of SepF depletion, time-lapse microscopy of HTQ239 cells on an agarose nutrition pad without tryptophan was performed. Over time the cell size strongly increased (Fig. 4d, Supplementary Movie 2). The cells failed to establish a cell division plane, resulting in bloated cells, which is a phenotype similar to what was observed before when both FtsZ homologs, FtsZ1 and FtsZ2, of *H. volcanii* were either mutated or depleted[34,35]. The increase of the cell size also explains why the optical density of the culture of the SepF depletion strain still slowly increased. The severe cell division defect after SepF depletion and the strong increase of the cell size imply an important role of SepF in the division process. Besides functioning as a possible membrane anchor for FtsZ1 or FtsZ2, SepF might also be involved in the regulation of cell division.

Additionally, now having the SepF depletion strain HTQ239 it was possible to assess the functionality of the SepF-GFP fusion protein. To that end, plasmid pSVA13504 was created allowing expression of SepF-GFP under the control of the native *sepF* promotor. When expressed in the wild type H26, SepF-GFP showed the same localization in medium with and without tryptophan as described in Fig. 1. Also, in the SepF depletion strain localization of SepF-GFP was observed (Supplementary Fig. 3a). Cells of HTQ239 expressing SepF-GFP were already elongated under non-SepF-depleting conditions compared to H26 and became even more filamentous under SepF depletion conditions (Supplementary Fig. 3b). Additionally, an increase of SepF-GFP foci was observed in the elongated cells (Supplementary Fig. 3a). Moreover, growth of HTQ239 + pSVA13505 was unaffected in the medium supplemented with tryptophan compared to the control and it seemed that expression of SepF-GFP complemented the growth defect observed in HTQ239 when grown in medium without tryptophan (Supplementary Fig. 3c). However, a spot dilution assay with HTQ239 + pSAV13504 on selective plates under non-inducing conditions showed that SepF-GFP is not fully functional as the viability of the cells remained as low as the control (HTQ239 + pTA1392) compared to H26 (Supplementary Fig. 3d). Furthermore, SepF-GFP does not interfere with the function of the wildtype SepF as H26 cells expressing SepF-GFP grew similar and showed the same viability as H26 without additional SepF-GFP expression (Supplementary Fig. 3c, d).

**Increased SepF levels in *H. volcanii* have no effect on growth and cell shape**. Several studies in bacteria reported that overexpression of *sepF* hampers cell division[20,21]. To study whether overproduction of the archaeal SepF also leads to impaired cell division, *sepF* was cloned in plasmid pTA1992 (resulting in pSVA5960). To quantify the amount of overexpression a similar plasmid was cloned that allowed the expression of SepF with a C-terminal HA-tag under control of the same promotor as *sepF* in pSVA5960. Production of SepF-HA from backbone plasmid pTA1992 was 1.5 fold higher compared to wild-type levels (Supplementary Fig. 4).

The investigated strains were first grown in medium with tryptophan and diluted later into selective medium without tryptophan while the $OD_{600}$ was constantly measured (Fig. 5).

As observed before, the SepF depletion strain stopped growing after dilution into medium without tryptophan. Additional expression of *sepF* in the wild-type strain had no effect on the growth as compared to H26 without *sepF* expression plasmid pSVA5960. Both strains grew in a similar manner in medium with and without tryptophan. Moreover, additional expression of

*sepF* in the *sepF* depletion strain HTQ239 complemented the growth defect caused by the SepF depletion (Fig. 5). To further analyze the effect of the increased *sepF* levels in H26 and HTQ239, cells were imaged every 3 h after dilution into medium without tryptophan (Fig. 6a) and the cell length (Fig. 6b) and area (Fig. 6c) were determined (the distribution of all cell shape data is provided in Supplementary Figs. 5 and 6).

Before dilution into medium without tryptophan all strains were rod-shaped with an average length ranging from 3 to 4 μm ($t = 0$). During the next 9 h cells of strain H26, H26 + pSVA5960 and HTQ239 + pSVA5960 slowly changed their cell shape from a rod-shaped to a plate-shaped form and the cell length was reduced to 3 μm as expected for normal growth of *H. volcanii*[39]. In contrast, the length of the depletion strain strongly changed within 9 h. Already after 3 h in medium without tryptophan HTQ239 cells showed an increased length compared to H26. Six hours after SepF depletion the cells had two times the length of H26 at the same time point and 3 h later cells had three times the length of H26 (Fig. 6b). Moreover, the cell shape of the depletion strain not only changed in length but also in width, resulting in a strongly increased cell size within 9 h. After 3, 6, and 9 h in medium without tryptophan, HTQ239 cells showed 2-, 5-, and 6-times larger cell area than H26 cells (Fig. 6c). After 24 h in medium without tryptophan, H26, H26 + pSVA5960, and HTQ239 + pSVA5960 were plate-shaped with an average length of 1.8 μm, the typical cell morphology for *H. volcanii* cells in the stationary phase[39]. The cells of HTQ239 without any plasmid were partially hugely bloated or disrupted and most of the cells had a cragged cell surface. Due to the different grey values, the HTQ239 cells had in the acquired images at the last time point it was no longer possible to measure the size with the image analysis software.

**Localization of FtsZ1-GFP and FtsZ2-GFP in SepF-depleted *H. volcanii* cells**. In order to study the effect of the SepF depletion on the localization of FtsZ1 and FtsZ2, strain HTQ239 was transformed with either a plasmid expressing *ftsZ1-gfp* (pSVA5910) or a plasmid expressing *ftsZ2-gfp* (pSVA5956) under the control of their own promotors. Expression of either *ftsZ1-gfp* or *ftsZ2-gfp* in the HTQ239 strain resulted in filamentation of the cells during SepF depletion (Supplementary Fig. 7a) similar to what was observed before (Supplementary Fig. 3b). Importantly, this filamentation phenotype is caused by the presence of a plasmid in the *sepF* depletion strain, and not due to the expression of GFP-tagged FtsZ1 and FtsZ2. Both plasmids for the expression of *ftsZ1-gfp* or *ftsZ2-gfp* are based on the backbone plasmid pTA1392. Transformation of that backbone plasmid in HTQ239 and subsequent depletion of SepF, resulted in filamentous cells as well while H26 transformed with pTA1392 maintained a normal cell shape (Supplementary Fig. 7). Why the presence of a plasmid in the *sepF* depletion strain causes the filamentation phenotype is currently unknown, but it has been observed previously that the presence of a plasmid in *H. volcanii* strains having mutations that are linked to cell shape or division are leading to a higher amount of rod-shaped to filamentous-shaped cells compared to cells without plasmid at the same growth stage[34,40]. Like before, cells were imaged every 3 h after SepF depletion. Over time, the cell length of both strains strongly increased compared to the wild type. Nine hours after dilution into medium without tryptophan HTQ239 expressing *ftsZ1-gfp* had an average cell length of 12.6 μm and HTQ239 expressing *ftsZ2-gfp* an average length of 13.8 μm, almost five times longer than H26 at the same time. Twenty-four hours after SepF depletion cells were still filamentous but without further increase of the cell length compared to the cells imaged 15 h before (Supplementary Fig. 6).

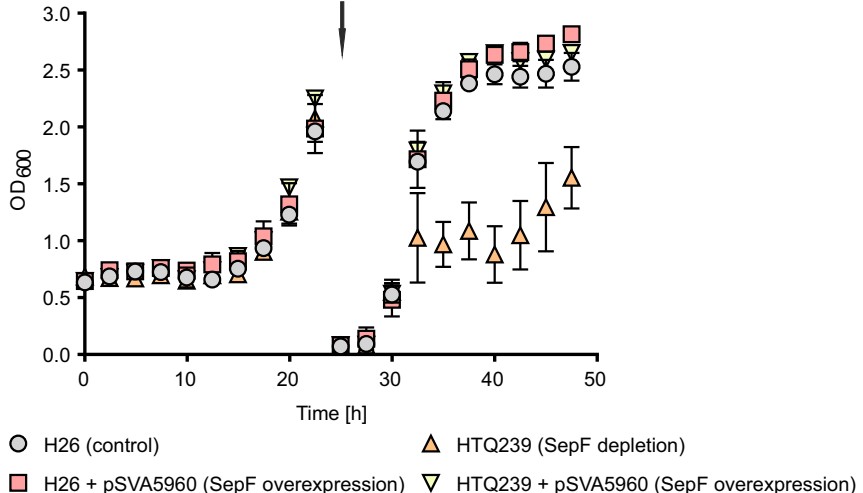

**Fig. 5 Growth curve of H26 and HTQ239 overexpressing SepF in the presence and absence of tryptophan.** Growth curve of H26 (gray), H26 + pSVA5960 (red), HTQ239 (orange) and HTQ239 + pSVA5960 (yellow) grown in CAB-medium supplemented with tryptophan and uracil. All four strains grew the same before dilution into CAB-medium without tryptophan (indicated by the arrow). After dilution, HTQ239 stopped growing while the other strains grew unaffectedly. Moreover, additional expression of SepF from plasmid pSVA5960 had no effect on the growth of H26 and even complemented the growth defect of HTQ239 in medium without tryptophan. The data for this growth curve was acquired from three independent cultures per strain. Error bars indicate the standard deviation.

Interestingly, depletion of SepF had a different effect on FtsZ1 localization compared to FtsZ2 localization. Before dilution into medium without tryptophan FtsZ1 localized to midcell in strain HTQ239 in almost all observed cells (Fig. 7a). Three hours after depletion the cells started to increase in length and most of the cells had one or two FtsZ1 rings. With increasing cell length ($t =$ 3 and $t = 9$) additional FtsZ1 rings were observed in the elongated cells. After 24 h of SepF depletion, cells with up to nine FtsZ1 rings, distributed over the cell body, were detected. During the depletion process the longer the cells became the more FtsZ1 rings were assembled in these filamentous cells (Fig. 7a). Moreover, the demographs from 9 and 24 h after dilution into medium without tryptophan showed that one FtsZ1 ring was always located close to one of the two cell poles (Supplementary Fig. 8a).

Before SepF depletion was induced, most of the HTQ239 cells expressing *ftsZ2-gfp* showed one FtsZ2 ring per cell located in the cell center (Fig. 7b, Supplementary Fig. 8b). In the next nine hours after SepF depletion, the cells became filamentous as well. Cells on average longer than 13 µm assembled two FtsZ2 rings while cells shorter than 13 µm had only one FtsZ2 ring. In contrast to the number of FtsZ1 rings observed in HTQ239 cells, a maximum of three FtsZ2 rings was observed in the most elongated cells. Moreover, in elongated cells ($t = 6–24$ h, >10 µm) with only one FtsZ2 ring, the ring was always located close to one cell pole. In cells with two FtsZ2 rings each ring was located near one of the two poles. The third FtsZ2 ring in cells that assembled three rings was located at midcell between the other two FtsZ2 rings (Supplementary Fig. 8b). Since SepF was shown to directly interact with FtsZ in bacteria a decrease of FtsZ1 and FtsZ2 rings after SepF depletion was expected for *H. volcanii*. Instead, the number increased to up to three rings per cell for FtsZ2 and up to nine rings for FtsZ1. It is possible that the remaining SepF after depletion was sufficient to anchor up to three FtsZ2 rings to the membrane. However, the high number of additional FtsZ1 rings in SepF-depleted cells might point to another, yet unknown, FtsZ1 membrane anchor instead of SepF. Moreover, besides filament formation of HTQ239 transformed with either of the two plasmids, there were slight differences in the cell morphology detectable between cells expressing FtsZ1-GFP or FtsZ2-GFP

(Supplementary Fig. 8c). HTQ239 cells expressing FtsZ2-GFP had an increased cell shape compared to cells that expressed FtsZ1-GFP.

**SepF from *H. volcanii* interacts with FtsZ2 but FtsZ1 is important for proper SepF localization.** To further investigate whether SepF directly interacts with FtsZ1 or FtsZ2, pulldowns from the cytosolic or membrane fraction of cross-linked SepF-HA expressing cells (HTQ236) were performed using magnetic beads coated with antibodies against the HA-tag. To ensure that the endogenous *ha*-tag on *sepF* has no influence on the cell growth and shape, strain HTQ236, was compared to the wild-type H26 (Supplementary Fig. 9). The elution fractions of the pulldowns were subsequently analyzed via Western blot analysis using specific antibodies against either FtsZ1 or FtsZ2, and by mass spectrometry. Mass spectrometry could not identify any peptides of FtsZ1 in the elution fractions of either the control sample from H26 or the SepF-HA expressing strain HTQ236 (Supplementary Table 4), whereas FtsZ2 was present in high amounts in the pulldown fractions from the HTQ236 strain. The same result was obtained using Western Blot analysis with α-FtsZ1 and α-FtsZ2 antibodies. Whereas only an unspecific signal was detected for the α-FtsZ1 antibody in all pulldown fractions (Fig. 8b), a very clear signal indicating the presence of FtsZ2 specifically in the pulldown fraction from the membranes of the HTQ236 strain (Fig. 8b, comprehensive gels and Western-Blots with load, flow-through, last wash fraction of the co-immunoprecipitation are provided in Supplementary Fig. 10). These pulldown assays thus demonstrated a direct interaction between SepF and FtsZ2.

Interestingly, though only interaction of SepF with FtsZ2 was detected, FtsZ1 needs to be present in the cell for correct SepF localization in *H. volcanii* while the absence of FtsZ2 does not impair proper foci formation of SepF-GFP. In an *ftsZ1* deletion strain SepF-GFP was not organized in ring-like structures but in large clusters and in an *ftsz1/ftsZ2* double knock-out strain SepF-GFP had a diffuse localization while in an *ftsZ2* deletion strain normal SepF foci formation was observed (Fig. 8c). Moreover, focusing on less deformed cells of the *ftsZ1* deletion strain with a constant width, it is apparent that SepF-GFP also forms clusters

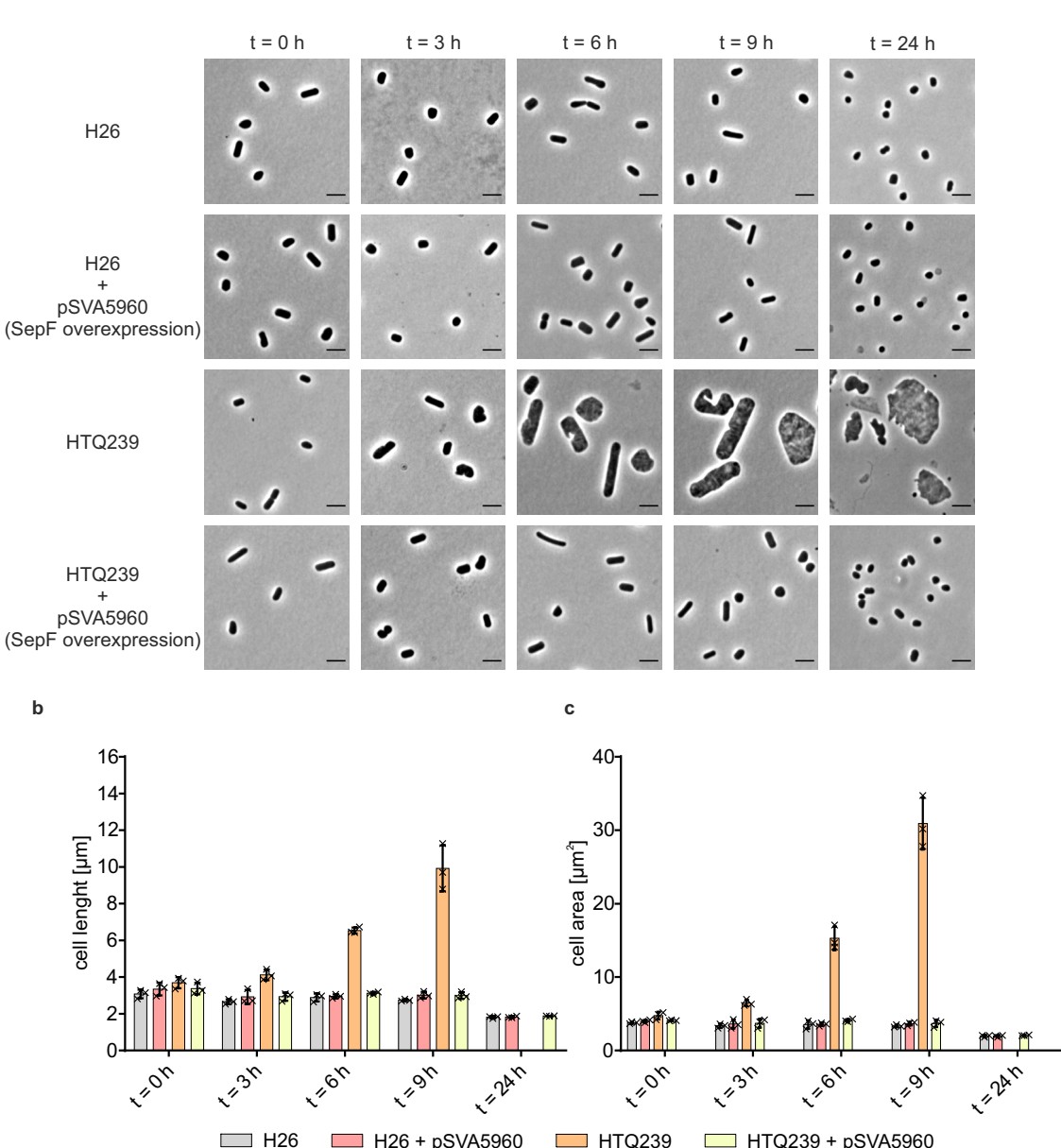

**Fig. 6 Cell shape analysis of H26 and HTQ239 during *sepF* overexpression in the absence of tryptophan. a** Microscopy of H26 and HTQ239 cells with or without SepF overexpression plasmid pSVA5960. Cells were imaged in 3 h steps after SepF depletion for 9 h and one last time 24 h after depletion. The experiment was repeated three times independently with similar results. Scale bar: 4 μm. **b** Corresponding mean cell length of strain H26 (gray), H26 + pSVA5960 (red), HTQ239 (orange), and HTQ239 + pSVA5960 (yellow) at different time points after SepF depletion. **c** Corresponding mean cell area at different time points after SepF depletion. Cell shape parameters were obtained from three independent experiments including the cell size of >1000 cells per strain and time point. Error bars indicate the standard deviation, each mean per replication is indicated by a x.

in these cells. Though, SepF is not organized in a defined focus (Supplementary Fig. 11), like observed in H26 or the *ftsZ2* deletion strain. It has been indicated previously that FtsZ1 is one of the first cell division proteins arriving at the future septum in *H. volcanii*[34]. According to that FtsZ1 recruits other, so far unknown, factors to the site of cell division that organize SepF into a dense, ring-like structure at the site of cell division. Moreover, the MTS of SepF fused to GFP did not show localization in the *ΔftsZ1*, *ΔftsZ2* strain or the double *ftsZ* knock-out (Fig. 8d) compared to the wild type H26 (Fig. 2c). In all three strains, diffuse localization of MTS$_{SepF}$-GFP was observed by fluorescence microscopy.

**H. volcanii SepF does not form oligomers in vitro**. Bacterial SepF was reported to form oligomers by lateral interaction between SepF dimers[15]. Since G109 of *B. subtilis* SepF, which in bacteria is important for this interaction, is not conserved in archaeal SepF homologs, it was tested whether *H. volcanii* SepF could also form oligomers. *H. volcanii* SepF was expressed in *E. coli* and purified by nickel affinity-chromatography (Supplementary Fig. 12a, b) and size exclusion chromatography experiments were performed. SepF eluted at a position corresponding to a molecular weight of ~30 kDa, corresponding to a dimeric protein and no protein eluted at elution volumes corresponding to higher oligomeric complexes (Fig. 9a). The eluted protein was

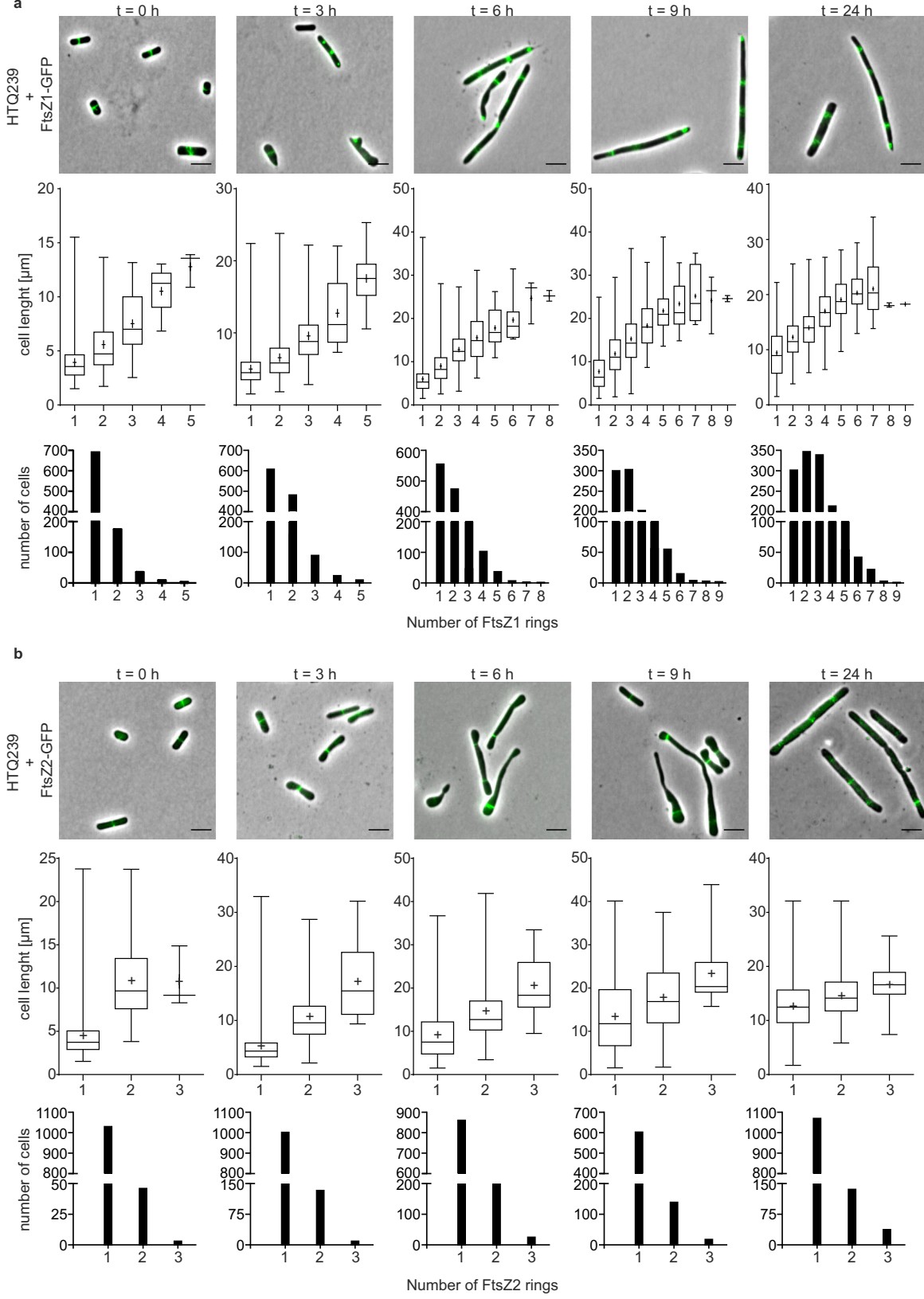

analyzed via Western blot using antibodies against the His-tag to verify that the eluted protein is SepF (Fig. 9b, whole Western blot image is provided in Supplementary Fig. 12c). Thus *H. volcanii* SepF forms a dimer and no indication of oligomerization could be found in vitro. Indicating that the archaeal SepF does not undergo polymerization as a similar result for the SepF protein of *M. smithii* was observed, which was even capable of membrane deformation[41].

**Fig. 7 Localization of FtsZ1 and FtsZ2 in SepF-depleted cells. a** First row: Fluorescence microscopy of HTQ239 cells additionally expressing *ftsZ1-gfp* under the control of its native promotor at different time points (0, 3, 6, 9, and 24 h) after SepF depletion. Middle row: Cell length in relation to the number of FtsZ1 rings assembled per cell. Over time the cells became filamentous and the number of FtsZ1 rings increased up to nine rings in the most elongated cells. Third row: Distribution of FtsZ1 rings amongst the analyzed cells at the indicated time points after SepF depletion. **b** First row: Localization of FtsZ2-GFP by fluorescence microscopy in HTQ239 at different time points after SepF depletion. Middle row: Cell length in relation to the number of FtsZ2 rings assembled per cell. Cells became filamentous over time and the number of FtsZ2 rings partially increased up to three rings per cell. Third row: Distribution of FtsZ2 rings amongst the analyzed cells at the indicated time points after SepF depletion. The plots summarize the results from three independent experiments per strain and timepoint after depletion. In the box plots, minimum and maximum values are indicated by the whiskers, the median is indicated by a horizontal line and the mean by a +. Scale bars: 4 μm.

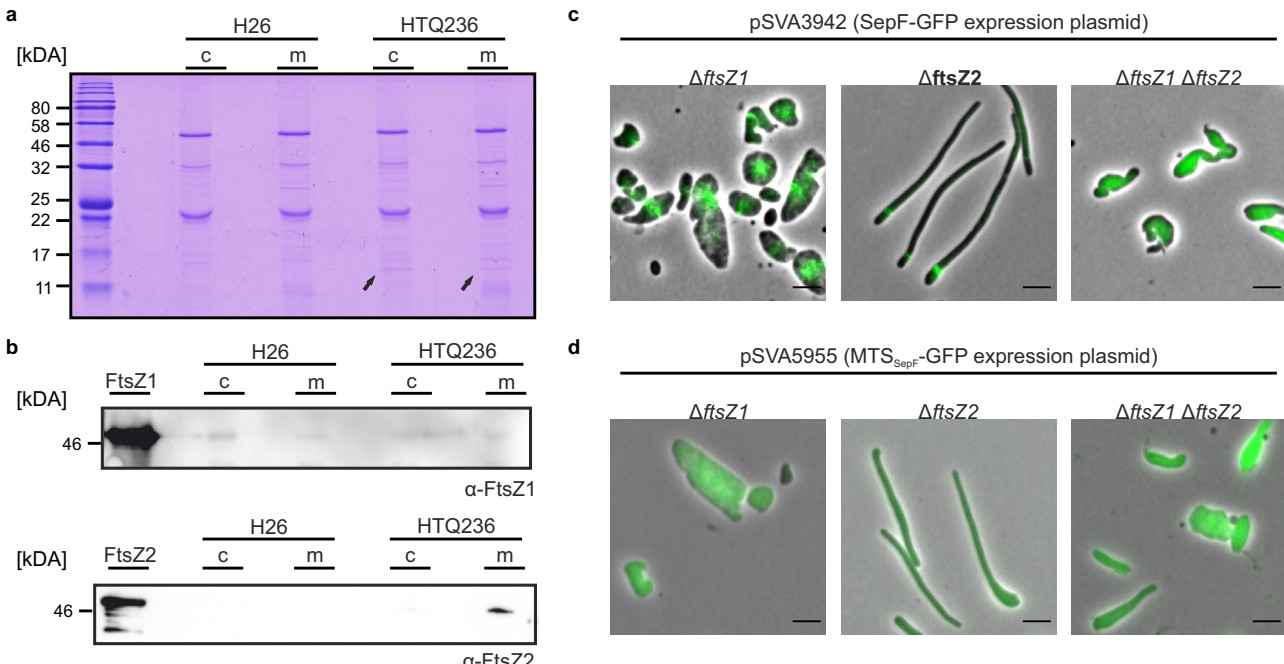

**Fig. 8 Immunoprecipitation of SepF-HA.** Eluted material of cytosolic (c) or membrane (m) fractions used for immunoprecipitation from H26 or HTQ236 (*sepF::sepF-ha*) were loaded on 15% SDS-gels **a** and analyzed via Western blots using antibodies either against FtsZ1 or FtsZ2 **b**. As control purified FtsZ1 or FtsZ2 were loaded. The HA-tagged SepF was visible on the Coomassie-stained gels in all pulldown fractions of strain HTQ236 (indicated by the black arrows). Experiments were repeated three times. **c** Fluorescence microscopy of SepF-GFP in *ftsZ1*, *ftsZ2*, and *ftsZ1/ftsZ2* deletion strains. **d** Fluorescence microscopy of MTS_SepF-GFP in *ftsZ1*, *ftsZ2,* and *ftsZ1/ftsZ2* deletion strains. The experiments were repeated three times independently with similar results. Scale bars: 4 μm.

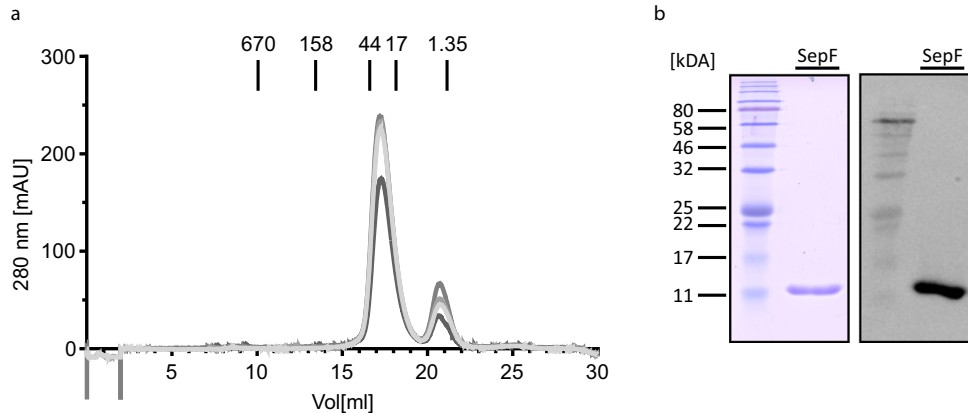

**Fig. 9 Size exclusion chromatography with SepF. a** Purified SepF was loaded on a Superdex 200 10/300 GL column. Absorption at 280 nm at the different elution volumes for four independent experiments are shown. Elution volumes of protein size markers are indicated in kDA. **b** SDS-gel and Western blot with anti-His antibodies of SepF after SEC. Elution fractions of two of the four repetitions were loaded on an SDS-gel (see Supplementary Fig. 12c).

## Discussion

SepF is an important protein involved in the cell division process of several bacterial species, including Cyanobacteria, Firmicutes, and Actinobacteria, functioning as an alternative membrane anchor for FtsZ[15,16]. Genes encoding SepF homologs are also found in FtsZ containing archaea[24,41]. Studies of archaeal SepF homologs were until now limited to the elucidation of the crystal structure of the putative SepF proteins from *A. fulgidus* and *P. furiosus*[15]. Very recently, the crystal structure of SepF of *Methanobrevibacter smithii* in its apo-form and in complex with the C-terminal domain of FtsZ was solved[41] demonstrating that SepF also directly interacts with FtsZ in archaea. Here we present, a functional study of an archaeal SepF protein and show that *H. volcanii* SepF is an essential component of the archaeal cell division machinery.

Fluorescence microscopy of SepF-GFP in *H. volcanii* showed localization in a ring-like structure at the cell division plane in the center of the cell, similar to the localization of bacterial SepF[21]. The same SepF localization pattern was also observed by immunofluorescence in the archaeon *M. smithii* with the slight difference that SepF already localized to the future sites of cell division before sister cells were segregated[41]. An N-terminal GFP fusion resulted in a diffuse localized GFP-SepF signal, suggesting GFP prevented the membrane localization of SepF. On the other hand, SepF with the C-terminal GFP tag is not fully functional as it did not fully complement SepF depletion. Similar to the SepFs from bacteria, SepF from *H. volcanii* contains an N-terminal membrane targeting site (MTS)[41]. Interestingly, the short MTS sequence of SepF fused to GFP, efficiently localized the fluorescent protein to midcell. Hence, besides anchoring SepF to the cell membrane, the MTS seems to be important for SepF localization at the site of cell division. One possibility for this localization is that the MTS of archaeal SepF is an additional interaction site of the SepF dimer, besides the β-sheets[15,41], resulting in the recruitment of $MTS_{SepF}$-GFP to the site of cell division by native SepF. However, localization of the $MTS_{SepF}$ in both *ftsZ1/2* deletion mutants and the double knock-out was diffuse implicating that the positioning of SepF to the site of cell division is supported by another factor. The presence of this protein is dependent on both tubulin homologs, FtsZ1 and FtsZ2 in contrast to SepF that showed normal localization in the *ΔftsZ2* strain. Time-lapse microscopy of SepF-GFP producing cells showed that SepF is localized at the future site of cell division immediately after closure of the septum, making SepF one of the first cell division proteins to arrive at the new septum. It is known that the future cell division plane in *H. volcanii* is assembled early during the cell cycle since FtsZ1 and FtsZ2 arrive shortly after the end of the previous round of cell division at the future cell division site[34,35]. However, the assembly hierarchy of SepF, FtsZ1, and FtsZ2 is unknown so far. We show here that FtsZ2 interacts with SepF, suggesting that SepF functions as a membrane anchor which brings FtsZ2 to the division site. In contrast, we were not able to show a direct interaction between SepF and FtsZ1. This leads to the assumption that SepF is an exclusive membrane anchor for FtsZ2 and that there is another, likely non-canonical, membrane anchor for FtsZ1 present in archaea that involve two FtsZ homologs for cell division. Moreover, FtsZ2 localization was reported to highly depend on the presence of FtsZ1[34]. Our SepF localization experiment in *ftsZ1* and *ftsZ2* deletion mutants showed that the positioning of SepF and its organization in a ring-like structure is also dependent on the presence of FtsZ1 in the cell. This indicates that FtsZ1 might be the first cell division protein that is placed at the future site of cell division in archaea that have two FtsZ homologs. Investigation on SepF from *Mycobacterium tuberculosis* showed that the positioning of SepF is dependent on FtsZ[21]. This was recently confirmed as the

interaction of SepF with FtsZ was crucial for its membrane binding in *C. glutamicum*[19]. As FtsZ1 does not interact with SepF, the presence of FtsZ1 most likely is important to recruit other factors to the site of cell division that brings SepF to the cell center and in turn leading to the recruitment of FtsZ2 to the site of cell division, resulting in the formation of a mature divisome. This comes along with the observation by Liao et al. that describes FtsZ1 as a recruitment hub for cell division proteins and FtsZ2 being involved in the constriction process[34] at which SepF might be the linker between both steps.

Our SepF depletion experiment showed that the lack of SepF had no negative effect on FtsZ1 ring formation as the number of additional FtsZ1 rings increased to up to nine rings during the late stages of depletion. However, only a maximum of three FtsZ2 rings was seen during SepF depletion. Possibly their formation was triggered by SepF remnants from before depletion. Interestingly, the presence of a plasmid in SepF depletion strain HTQ239 resulted in filamentous cells, in contrast to the swollen HTQ239 cells that were observed under SepF depletion conditions when grown devoid of a plasmid in uracil supplemented selective media. The reason for the filament formation of HTQ239 cells containing a plasmid is currently unknown but similar results have been observed when other cell division-related proteins of *H. volcanii* were perturbed and the mutants contained plasmids[34]. Additionally, it has been reported before that the presence of a plasmid supports rod formation in the auxotrophic *H. volcanii* laboratory strain[40,42].

The swollen HTQ239 cells resembled FtsZ depleted *Staphylococcus aureus* in which, as a result of FtsZ depletion, one of the major cell wall synthetic enzymes was displaced[43]. From bacteria there is evidence that the SepF protein is also involved in later steps of cell division, indicating that SepF also has a regulatory role during cytokinesis[14,16,19–21,44–46]. Therefore, SepF needs to be tightly regulated as changes in native SepF levels lead to severe cell division defects. Especially overexpression of SepF was shown to completely block cell division in different bacteria[19–21]. Interestingly, additional overproduction of FtsZ mitigated the effect of SepF overexpression, implicating that the SepF/FtsZ ratio and the reduction of freely diffusing SepF are crucial for correct cell division in bacteria[20]. In contrast, no negative effect was observed in *H. volcanii* cells that additionally expressed SepF from a plasmid with a constitutive promotor. Moreover, SepF expression from plasmid pSVA5960 rescued the SepF-depleted cells. The exact intracellular concentrations of SepF and FtsZ2 are unknown but the ratio of both proteins apparently does not need to be as tightly regulated as in bacteria since the additional SepF expression and thereby the increased concentration of free archaeal SepF does not block cell division like observed for bacteria[20]. However, we must point out that the native SepF levels plus the 1.5-fold increased SepF expression from the plasmid might not be enough to actually block cell division in *H. volcanii*. In the bacterial studies on the effect of SepF overproduction, the levels of overexpression were not quantified, making a direct comparison difficult.

Nevertheless, we assume that SepF has also a direct or indirect regulatory role in *H. volcanii* as during SepF depletion constriction of the formed FtsZ1 and FtsZ2 rings were blocked and no septum closure had been observed. The force for septum closure in bacteria is generated by incorporation of new cell wall material at the septum[3–5] and free SepF in *B. subtilis* was reported to lead to delocalization of proteins of the septal PG synthesis machinery[20]. Moreover, in *M. tuberculosis* SepF was reported to interact with MurG, a protein involved in the maturation of the PG precursor Lipid-II[45]. In another actinobacterium, *C. glutamicum*, no septal PG incorporation has been observed after SepF depletion[19]. During SepF depletion in *H. volcanii* we observed a

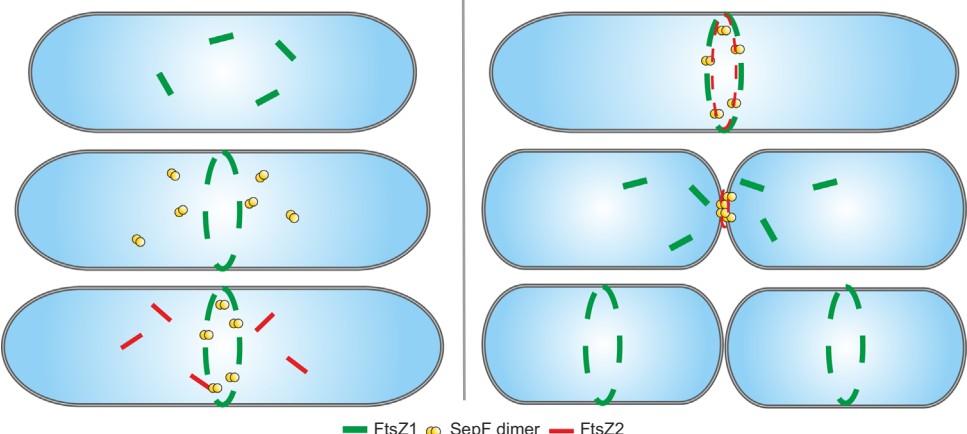

**Fig. 10 Model of cell division in *H. volcanii*.** Left: Directly after the last round of cell division FtsZ1 (green) is positioned at the future site of cell division, by a so-far unidentified mechanism, forming the FtsZ1 ring. Next, SepF dimers (yellow) are localized in an FtsZ1-dependent manner to the site of cell division, and SepF recruits and anchors FtsZ2 (red) to the septum, resulting in the formation of the FtsZ2 ring. Right: After assembly of the mature divisome constriction is initiated either by force generation or incorporation of new cell wall material. From the septum released FtsZ1 proteins then directly localize to the future site of cell division.

strong increase in the cell size, most likely the result of uncoordinated incorporation of new cell wall material all over the cell body. This suggests that upon disruption of the archaeal SepF ring incorporation of new cell wall material is perturbed. It is possible that the presence of SepF at the septum is important for the correct coordination of the cell wall synthesis machinery to the site of cell division in archaea. Indeed, previous studies reported that in several archaeal species new S-Layer is incorporated at the cell center[47–49].

Based on our study on the archaeal SepF protein and the previously reported study on both FtsZ homologs in *H. volcanii*[34], we propose the following model for divisome formation in Euryarchaeota (Fig. 10): Directly after cell division, FtsZ1 is localized to the future site of cell division by a so-far unidentified regulator. Subsequently, FtsZ1 protofilaments are attached to the membrane by an unknown non-canonical membrane anchor forming the Z1-ring. SepF is recruited to the cell center by a transient interaction with FtsZ1 or another factor that's localization to the site of cell division is dependent on FtsZ1. SepF dimers in turn tether FtsZ2 filaments to the membrane resulting in the formation of the second Z-ring and the completion of the early divisome. The formed Z2-ring then provides a scaffold for further downstream cell division proteins that are involved in septum closure either by the generation of force or the incorporation of new cell wall material. We are aware that this is a simplified model as we expect many more proteins to be part of the euryarchaeal divisome that yet have to be identified. Moreover, it is likely that only the proteins of the early divisome like FtsZ1, FtsZ2, and SepF are conserved in Euryarchaeota as the cell envelopes differ strongly amongst euryarchaeal species[50,51]. Hence, it can be assumed that proteins for late steps in cell division are not conserved amongst Euryarchaeota as also the machinery for cell wall material incorporation at the septum need to be adapted to the different types of cell envelopes.

In conclusion, the predicted archaeal SepF protein is an essential part of the euryarchaeal divisome that not only provides a membrane anchor for FstZ2 but might also be important for septum closure.

## Methods

Unless otherwise stated all chemicals and reagents were obtained either from Carl Roth or Sigma-Aldrich.

**Strains and growth media**. *H. volcanii* strains were either grown in rich medium (YPC-medium) for transformations or selective CA-medium[52] supplemented with an expanded trace element solution (CAB-medium)[35] for experiments. The strain with the hemagglutinin (HA)-tag on SepF, HTQ236, and H26 required the addition of 0.45 mM uracil when grown in CAB-medium. The SepF depletion strain HTQ239 needed the addition of 0.25 mM tryptophan for normal growth in CAB-medium and 0.45 mM uracil. To deplete SepF, strain HTQ239 was grown in CAB-medium in the absence of tryptophan. Cultures smaller than 5 ml were grown in 15 ml culture tubes rotating at 45 °C while larger cultures were grown in flasks on shakers set to 120 rpm at 45 °C. Plates for the growth of *H. volcanii* on solid medium after transformation were prepared as described previously by Allers et al.[52]. Plates were incubated in plastic boxes to prevent evaporation at 45 °C. Growth curves of 20 ml cultures were automatically measured using the cell growth quantifier (CGQ, Software (CGQuant 7.4)) from Aquila Biolabs GmbH at 45 °C and constant shaking at 120 rpm.

*E. coli* strains were grown in LB-medium[53], supplemented with appropriate antibiotics (100 µg/ml ampicillin; 25 µg/ml kanamycin; 30 µg/ml chloramphenicol) if necessary, at 37 °C under constant shaking.

Archaeal and bacterial strains used in this study are listed in Supplementary Table 1.

**Bioinformatic analysis of the putative archaeal SepF protein**. The map of the genetic neighborhood of hvo_0392 was drawn with Gene Graphics[54] with a region size of 6500 base pairs (bp). To obtain a general overview of the archaeal SepF proteins, sequences from selected bacterial SepF proteins were aligned with the putative SepF proteins from different archaea, using the MUSCLE (MUltiple Sequence Comparison by Log-Expectation[55] web service with default settings accessed via Jalview (Version 2.11.0))[56]. The schematic overview of the putative SepF protein of *H. volcanii* was drawn with the Illustrator for Biological Sequences (IBS, Version 1.0)[57]. Amino acids with a conservation score >7 in the alignment were indicated in the protein scheme.

**Plasmid construction**. Enzymes for the amplification of inserts by polymerase chain reaction (PCR) (using PHUSION® polymerase), digestion, and ligation were obtained from New England Biolabs (NEB). Plasmids for expression of fluorescently tagged proteins in *H. volcanii* and homologous protein expression were constructed via classical restriction enzyme-based cloning into plasmid pIDJL-40, pTA1392, or pTA1992 for single protein expression and pSVA3943 for double expression, following the manufacturer's protocols. Inserts were amplified from genomic DNA of strain H26, isolated as described before[36]. Genes were under the control of a tryptophan inducible promotor p.tnaA[58] or a synthetic constitutive promotor (p.syn) in the pTA1991 backbone[59].

For the generation of the plasmid to add an HA-tag to the genomically encoded *sepF* (hvo_0392) at its C-terminus two fragments were amplified: A fragment ~500 bp upstream of the *sepF* stop-codon with a 5′ KpnI restriction site and 3′ *ha* sequence with a stop codon and a second fragment ~500 bp downstream of the *sepF* stop-codon with the complementary *ha*+ stop codon sequence at 5′ end and an XbaI restriction site at the 3′ end. Both fragments were assembled via overlap PCR using the forward primer of the upstream fragment and the reverse primer of the downstream fragment. The generated fragment was subsequently cloned into integrative plasmid pTA131.

The *sepF* knock-out plasmid was created by amplification of ~500 bp of the upstream region and ~500 bp of the downstream region of *sepF*. Both PCR products were linked together by digestion with BamHI and subsequent ligation. The fragment was then cloned into plasmid pTA131 via KpnI and XbaI.

To generate the plasmid for the construction of the *sepF* depletion strain, the gene *sepF* was cloned in pTA1369, resulting in tryptophan-inducible *sepF*. The gene construct was subsequently cut out from pTA1369 with BglII and cloned in between the upstream and downstream regions of *sepF* in its knock-out plasmid opened with BamHI.

The plasmids for heterologous protein expression in *E. coli* were created with the FX-cloning system described by Geertsma and Dutzler[60]. The sequences for the forward and the reverse primers to amplify *sepF*, *ftsZ1* (*hvo_0717*) and *ftsZ2* (*hvo_0581*) for FX-cloning were obtained from https://www.fxcloning.org/.

Primers used for the amplification of the different DNA fragments and the respective restriction enzymes for cloning are indicated in Supplementary Table 3. Plasmids used in this study are listed in Supplementary Table 2.

### Transformation of plasmids and genetic manipulation of *H. volcanii*. *H. volcanii*

was transformed using polyethylene glycol 600 (PEG600) as described before[36]. The selection for successful transformation was based on the complementation of the uracil auxotrophy of the used strains by the transformed plasmids. In brief: When 10 ml of culture reached an optical density (OD$_{600}$) of 0.8, cells were harvested (3000 × *g* for 8 min) and spheroblasts were formed by resuspending in buffered spheroblasting solution (1 M NaCl, 27 mM KCl, 50 mM Tris–HCl (pH 8) and 15% [w/v] sucrose) with 50 mM EDTA (pH 8). The resuspended cells were incubated at room temperature (RT) for 10 min. In the meantime, 1 µg of a respective deme-thylated plasmid, passed through a dam⁻/dcm⁻ *E. coli* strain (NEB) before, was mixed with 83 mM EDTA and unbuffered spheroblasting-solution (1 M NaCl, 27 mM KCl, and 15% [w/v] sucrose) to a final volume of 30 µl. DNA was added to the spheroblasts and gently mixed. After 5 min of incubation an equal amount of 60% PEG 600 was added, gently mixed, and incubated for 30 min. Subsequently, 1.5 ml spheroblast-dilution solution (23% salt water, 15% [w/v] sucrose, and 3.75 mM CaCl$_2$) was added, the tubes were inverted and incubated for an additional 2 min. After spheroblasts were harvested at 3000 × *g* for 8 min, 1 ml regeneration solution (18% saltwater, 1× YPC, 15% [w/v] sucrose, and 3 mM CaCl$_2$) was added and the pellet transferred to a sterile 5 ml tube. The pellet was incubated standing for 1.5 h at 45 °C before it was resuspended by gentle shaking and additional incubation for 3.5 h rotating at 45 °C. Transformed cells were then harvested (3000 × *g* for 8 min) and resuspended in 1 ml transformant-dilution solution (18% salt water, 15% [w/v] sucrose, 3 mM CaCl$_2$). At last, 100 µl of 10⁰, 10⁻¹, and 10⁻² dilutions of the transformed cells were plated on selective plates.

### Integration of a genomic HA-tag at the *sepF* locus. For the integration of an

endogenous HA-tag at *sepF* the pop-in/pop-out method was used[61]. H26 was transformed with integrative plasmid pSVA3947 and subsequently grown on selective CA-plates until colonies were visible. To induce the pop-out, one trans-formant colony was transferred to 5 ml non-selective medium (YPC-medium) and grown at 45 °C. When the culture reached OD$_{600}$ 1 it was diluted 1:500 into fresh YPC-medium. This was repeated three times in total. To select for successful pop-out events, 100 µl of the 10⁻² diluted pop-out culture was plated on 50 µg/ml 5-fluoorotic acid (5-FOA) containing CA-plates. Colonies from the 5-FOA plate were screened via colony PCR for successful HA-tag integration. For the colony PCR material from single colonies was transferred into 300 µl dH$_2$O and incubated for 10 min at 75 °C under constant shaking for complete cell lysis. Subsequently, 1 µl of the lysed cells were used as a template, and the PCR product was sent for sequencing.

### Construction of a SepF depletion strain. To exchange the native promotor of

*sepF* with a tryptophan-inducible promotor, integrative plasmid pSVA3954 was transformed in *H. volcanii* strain H98. Transformants were checked for correct upstream integration and orientation of the construct by colony PCR. A sub-sequent pop-out procedure was performed as described before. To screen for the promotor exchange, pop-out colonies were first re-streaked on selective plates containing tryptophan and uracil. After colonies had formed, they were transferred to a selective plate with uracil but without tryptophan. Colonies that were not able to grow without tryptophan were checked via colony PCR and subsequent sequencing for successful promotor exchange.

### Spot survival assays. To assess the viability of strain H26 and HTQ239 they were

grown in 20 ml Cab-medium supplemented with 0.45 mM uracil and 0.25 mM tryptophan at 45 °C under constant shaking to an OD$_{600}$ of 0.1. Two ml of each culture were harvested at 3000 × *g* for 8 min at room temperature and the pellet resuspended to a theoretical OD$_{600}$ of 0.2. A serial dilution ending at 10⁻⁵ was prepared and 5 µl of each dilution per strain was dropped on Ca-plates either complemented with 0.45 mM uracil and 0.25 mM tryptophan or complemented with 0.45 ml uracil. The plates were incubated for 2 days at 45 °C in sealed plastic bags. The rest of the harvested culture was re-inoculated in 20 ml Cab-medium complemented only with 0.45 mM uracil to repeat the spot survival assay with cells grown under SepF-depleting conditions.

Spot dilution assays with H26 and HTQ239 containing plasmid pSVA13504 were done the same with the exception that no uracil had to be added to the selective medium. Moreover, the final dilution was 10⁻⁴.

### Microscopy. To investigate the cellular localization of the putative archaeal SepF

protein alone or in combination with FtsZ1 or FtsZ2 in different background strains, fluorescence microscopy was used. To image *H. volcanii*, a 3 µl sample of early exponentially growing cells (OD$_{600}$ < 0.1) was spotted on a 0.3% [w/v] agarose pad containing 18% buffered salt water (144 g/l NaCl, 18 g/l MgCl$_2$·6H$_2$O, 21 g/l MgSO$_4$·7H$_2$O, 4.2 g/l KCl, 12 mM Tris/HCl, pH 7.5). After the sample was dried on the pad it was covered with a cover slide and observed with an inverted microscope (Zeiss Axio Observer.Z1, controlled via VisiView (Version (4.5.0.6.) software) equipped with a temperature-controlled chamber at 45 °C.

For over-night time lapse microscopy, an agarose pad containing nutrients was prepared by mixing 0.3% agarose [w/v] with CAB-medium supplemented with 5% [w/v] sucrose. One ml of the nutrition pad solution was poured in a round Delta T Dish (Bioptechs Inc.). After the pad had solidified 3 µl of cells in very early exponential phase (OD$_{600}$ 0.05–0.1) were dropped on the pad. After the spots were dried the whole pad was flipped up-side down into another Delta T Dish and the dish was closed with a lid (Bioptechs Inc.) to avoid evaporation. Cells were imaged at 45 °C over-night with image acquisition every 15 or 30 min for 16 h. For the observation of microcolonies, cells were grown over night on agarose pads containing nutrients at 45 °C and were imaged the next morning.

To image strain HTQ239 transformed with different plasmids, precultures of the respective transformants were grown in 30 ml CAB-medium supplemented with tryptophan to an OD$_{600}$ of 0.01. Cells were subsequently harvested at 3000 × *g* for 20 min in a centrifuge heated to 45 °C. The supernatant was discarded, and the cells resuspended in pre-warmed CAB-medium and grown again at 45 °C under constant shaking. Cells were imaged every 3 h for 9 h and one last time 24 h after growing in media without tryptophan. Images were analyzed using FIJI (Version 1.51)[62] and the MicrobeJ plug-in (Version 5.13 l)[63].

### Heterologous expression and purification of SepF-His, FtsZ1-His, and FtsZ2-Strep. The different proteins were produced by heterologous expression in the *E.

coli* Rosetta™ strain. For overexpression, Rosetta™ cells were transformed with plasmid pSVA3969 (SepF-His), pSVA3970 (FtsZ1-His), or pSVA5912 (FtsZ2-Strep) and grown in 1 l LB-medium supplemented with kanamycin (25 µg/ml) and chloramphenicol (30 µg/ml) at 37 °C under constant shaking. After reaching an OD$_{600}$ of 0.5 expression was induced by adding 0.5 mM β-D-1-thiogalactopyr-anoside (IPTG) (Fisher Science) and cells were grown for additional 3 h at 37 °C. Subsequently, cells were harvested by centrifugation at 6000 × *g* for 20 min. The pellet was immediately frozen in liquid nitrogen and stored at −80 °C until used.

For purification of His-tagged proteins the frozen pellet was thawed and resuspended in 20 ml Buffer A (1.5 M KCl, 100 mM NaCl, 50 mM L4-(2-hydroxyethyl)-1-piperazineethanesulfonic acid (HEPES), 1 mM dithiothreitol (DTT), pH 7.5) supplemented with 10 µg/ml DNase I (Roche) and cOmplete™ EDTA-free Protease Inhibitor (Roche). Cells that produced SepF-His were opened using a Microfluidizer®, cells containing overproduced FtsZ1-His were lysed by sonication (Sonopuls, Bandelin; Probe KE76; 35% power, 5 s pulse for 10 min). Cell debris were then removed by centrifugation at 3000 × *g* for 15 min at 4 °C and the supernatant used for a second centrifugation step at 26,000 × *g* for 20 min at 4 °C. The soluble fraction of SepF-His or FtsZ1-His was manually loaded on a, with Buffer A pre-equilibrated, 5 ml HisTrap™ HP column (GE Healthcare). For further purification steps the column was connected to a liquid chromatography system (Azura, Knauer). The column was washed in three steps with 10 ml Buffer A containing 10, 20, and 30 mM imidazole, respectively. The protein was eluted from the column via an imidazole gradient from 30 to 400 mM imidazole in Buffer A and collected in 2 ml fractions. For the purification of FtsZ2-Strep, cells were lysed as described for FtsZ1-His. The soluble fraction was loaded on a 3 ml gravity column of Strep-Tactin® Sepharose® (iba-lifesciences). Subsequently the column was washed with 15 ml Buffer A. To elute the protein 3 ml of Buffer A supplemented with 2.5 mM desthiobiotin was added and the flow-through collected in 1 ml fractions. FtsZ1-His and FtsZ2-Strep were concentrated by ultrafiltration (Amicon® Ultra 4 ml, 3 kDA cut-off) and subsequently frozen in liquid nitrogen and stored at −80 °C. The purified SepF-His was used for size exclusion chromatography (see: Determination of the oligomeric state of the archaeal SepF) while purified FtsZ1 (130 µM) and FtsZ2 (62 µM) were used as positive controls for the immunoprecipitation experiments with HA-tagged SepF.

### Immunoprecipitation of HA-tagged SepF. *H. volcanii* strain HTQ236 and H26 as

control were grown in 333 ml CAB-medium supplemented with uracil to an OD$_{600}$ of 0.1. The cells were harvested at 3000 × *g* for 20 min and subsequently resus-pended in 30 ml 18% salt water buffered with 10 mM HEPES. For crosslinking 1.2% [v/v] formaldehyde was added, and the cells incubated for 10 min at RT under constant shaking. To quench the crosslinking reaction 20 ml of 1.25 M glycine dissolved in 18% salt water were added and the cells harvested as described before. Following, the cells were washed two times with 20 ml ice cold 18% salt water with 1.25 M glycine. After the last washing step cells were resuspended in 3 ml 1×

phosphate-buffered saline (PBS) (137 mM NaCl, 2.7 mM KCl, 10 mM $Na_2HPO_4$, 1.8 mM $KH_2PO_4$, adjusted to pH 7.2) containing additionally 10 mM $MgCl_2$ and 10 µg/ml DNase I (Roche) and lysed via sonication (Sonopuls, Bandelin; Probe MS73; 35% power, 5 s pulse 5 s pause for 5 min). The lysed cells were then centrifuged at $3000 \times g$ and 4 ℃ for 15 min to remove cell debris. The supernatant was used for ultra-centrifugation to separate the membrane and cytosolic fraction: The samples were centrifuged at $200,000 \times g$ and 4 ℃ for 2 h. Subsequently, the membrane fraction was resuspended in 1× PBS supplemented with 1% Triton-X [v/v] with a volume equal to the volume of the supernatant. Both fractions were used for immunoprecipitation. The HA-tagged proteins were captured with the Pierce™ HA-Tag Magnetic IP/Co-IP Kit (Thermo Scientific™). Magnetic beads (0.25 mg) were prepared according to the manufacturer's protocol. The membrane and the cytosolic fractions, respectively, were added in 1 ml steps to the beads. Each loading step was followed by 30 min incubation at 4 ℃ while slowly rotating. The washing steps and the elution from the beads with non-reducing sample buffer were executed as described in the manufacturer's protocol.

Elution fractions of the pulldowns were either analyzed by mass spectrometry (MS) or Western blot analysis. For Western blotting the respective samples were separated by sodium dodecyl sulfate–polyacrylamide gel electrophoresis (SDS–PAGE). Gels were stained with Fairbanks (25% [v/v] isopropanol, 10% [v/v] acetic acid, 0.05% [w/v] Coomassie R) staining solution or blotted on polyvinylidenfluorid (PVDF) membranes using the Trans-Blot Turbo Transfer System (BioRad). As positive controls purified FtsZ1 or FtsZ2 was loaded on the respective gel. Each antibody against FtsZ1 or FtsZ2 (received from Iain Duggin, Sydney) raised in rabbits was incubated with the membrane over-night in a 1:1000 dilution (in 1× PBS + 0.1% Tween-20) at 4 ℃. Anti-HA antibodies (Sigma-Aldrich) were incubated over-night on the membrane in a 1:10,000 dilution as well. Subsequently, the membrane was incubated with the secondary antibody against rabbit coupled with a horseradish peroxidase (HRP) (Invitrogen) for 1 h in a 1:10,000 dilution. Signals from the Western blots were acquired with the iBright FL1500 imaging system (Invitrogen) after adding substrate for the HRP (Clarity Max Western ECL Substrate, BioRad) to the membrane. The MS sample preparation, measurement and data evaluation were performed by the Core Facility Proteomic of the ZBSA (Zentrum für Biosystemanalyse, Freiburg, Germany).

**Quantification of SepF overexpression**. To determine the degree of overexpression of *sepF* under the control of the synthetic promotor in plasmid pSVA5960, plasmid pSVA13505 was cloned. This plasmid allowed the expression of SepF-HA under the control of the synthetic promotor present in pSVA5960. As a control strain HTQ236 was used that allowed to follow the native SepF expression levels. Strain H26 was transformed with plasmid pSVA13505 and strain HTQ236 with pTA1392. The cells were grown to an $OD_{600}$ of 0.1 in 20 ml selective Cab-medium at 45 ℃ under constant shaking. Cells were harvested at $3000 \times g$ for 20 min at 4 ℃, the pellets resuspended in Cab-medium to a theoretical $OD_{600}$ of 10 and subsequently transferred to 1.5 ml reaction tubes. For cell lysis cell were harvested at $3000 \times g$ for 8 min at 4 ℃. The supernatant was removed, and the pellets resuspended in the same volume of 1× PBS supplemented with 2.5 mM $MgCl_2$ and 10 µg/ml DNase I. Then, n-dodecyl β-D-maltoside (DDM) to a final concentration of 0.1% was added and the cells incubated for 10 min on ice. To remove cell debris cells were centrifuged at $3000 \times g$ for 10 min and the supernatant used for analysis on 15% SDS-gels. After boiling for 10 min 10 µl per sample were loaded. Western-blotting was performed as described before. The first antibody anti-HA (1:10,000, Sigma-Aldrich) produced in rabbit was incubated on the membrane over-night at 4 ℃ under constant shaking. The second antibody anti-rabbit HRP coupled (1:10,000, Sigma-Aldrich) was incubated for 3 h at room temperature under constant shaking. Western blot signals were acquired as described before.

Signals were analyzed using Fiji (Version 1.51) and unspecific bands that appeared on the Western-blots were used as loading controls to normalize the SepF-HA signal per lane.

**Determination of the oligomeric state of the archaeal SepF**. To determine the oligomeric state of SepF, 500 µl of freshly purified protein was loaded on a Superdex 200 10/300 GL column (GE Healthcare) equilibrated with Buffer A (1.5 M KCl, 100 mM NaCl, 50 mM HEPES, 1 mM DTT, pH 7.5). Data were collected (PurityChrom® 5.09.036) for 60 min at a constant flow-rate of 0.5 ml/min. Additionally, a gel filtration standard (#1511901, BioRad) was run under the same conditions as SepF. The eluted protein was concentrated by ultrafiltration (Amicon® Ultra 4 ml, 3 kDA cut-off) and subsequently analyzed by SDS-gel electrophoresis and Western blotting. The HRP-coupled anti-His antibodies (Abcam®) for the detection of the His-tag on SepF were incubated with the membrane overnight in a 1:10,000 dilution.

**Plots and figures**. Raw data were imported in GraphPad Prism (Version 6.05) for the generation of plots. The model and final figures were assembled with CorelDraw X5.

**Reporting summary**. Further information on research design is available in the Nature Research Reporting Summary linked to this article.

## Data availability
The data that support the findings of this study are available from the corresponding author upon request. Source data are provided with this paper.

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

## Acknowledgements

We acknowledge support from "Zentrales Innovationsprogramm Mittelstand (ZIM) des Bundesministeriums für Wirtschaft und Energie (BMWi) (grant number ZF4653901AJ8) and VW Momentum (Grant number 94933) for P.N. The antibodies against FtsZ1 and FtsZ2 were a gift from Iain Duggin, UTS, Australia.

## Author contributions
P.N. conceived the initial idea and designed the experiments. Most of the experiments were performed by P.N. Under supervision of P.N., M.G. created strain HTQ236, plasmid pSVA3942, 3951, 3968, 3969 and 3970; M.D. created plasmid pSVA5910, 5912; C.E. created plasmid pSVA5954, 5956 and 5956. P.N. analyzed data, prepared figures/tables and wrote the manuscript. S.-V.A. reviewed drafts of the paper and acquired funding.

## Funding

## Competing interests
The authors declare no competing interests.
