## [Peer Review File · Nature Communications]

REVIEWER COMMENTS

Reviewer #1 (Remarks to the Author):

Nußbaum and co-workers did a great job revising the previous version of the manuscript. Most of my following comments are relevant but easy to correct. There is only one point that I strongly suggest to be revised, that is the interpretation of FtsZ1-dependent recruitment of SepF. I still believe authors should do a better job making clear that transient interactions, even though are possible, do not explain recruitment.

Specific comments:

The organization of the genomic neighborhood of *hvo_0392* it suggests that these genes are not organized in an operon (see Figure S1). There is a substantial gap of 120 Bp between *hvo_0392* and the downstream ORF (*hvo_0393*). Moreover, transcriptome data which were shared with our group implicate indeed that this is not an operon.

I agree that the appearance is suggestive, but we should be careful with our human common sense as nature is not always ruled by the pattern our eyes are used to see. Transcripts are a great example of how even the most experienced molecular biologists can be wrong without supportive data. I had a lot of fun looking at the literature after datasets, and there are not one but 2 RNAseq datasets that suggest exactly the opposite of the present claims: Laass 2019 (Plos One) and Gelsinger 2018 (JBac). Laas and colleagues offer a spreadsheet with detected transcript lengths and *hvo_0392* and *hvo_0393* shown as a single transcript. Gelsinger dataset is not so obvious but the dataset can be downloaded from NCBI and the coding transcript length is identical to Laass. It's an interesting observation that a putative DNA repair ATPase is co-transcribed with SepF. Please clarify this in the text.

replicon	transcript No.	strand	start	stop	length	RNA class	locus	Operon
CHR	330	+	347034	350568	3535	cd	0392	2

We agree and have performed CFU and spot dilution assays, which are now shown in Figure 4.

I am happy with the results, but I suggest authors adjust contrast/brightness of the image (and maybe deplete background) just to make colonies more visible.

We have observed that it makes a large difference how growth curves are performed with *H. volcanii*. Liao et al. 2020 performed their growth curves in 150 µl media in micro-titer plates. From our experience this method of growth curve acquisition does not work reliably for *H. volcanii* and differences in growth often won't be observed. For that reason, we used at least 15 ml of culture to acquire our growth curves. From their images of the different FtsZ deletions we would assume that the growth curve would look like the one for the SepF depletion strain if it would have been acquired in at least 15 ml. As suggested, we performed viability assays with the SepF depletion strain to clarify this point (See Fig 4).

Thank you. That makes completely sense. I understand how sensitive data can be to different experimental designs.

We have performed the recommended experiment and have added it to the manuscript. It shows that the localization of SepF is dependent on the presence of FtsZ1 (see pictures and Figure 8).

This is not fully supported by Fig 8. Please, note that Δ ftsZ1 and Δ ftsZ1 Δ ftsZ2 cells are deformed, with abnormal aspect ratios. Meanwhile, the presented Δ ftsZ2 cells are elongated rods, maintaining constant width. It is not possible to conclude from this data whether SepF is delocalizing due to morphological defects caused by the lack of FtsZ1 or because SepF is recruited by FtsZ1 itself. In fact, if you observe cells that are not as deformed, SepF is still capable of localization (Fig. 8C).

This strongly suggests that loss of shape control precedes SepF delocalization. Furthermore, the delocalization pattern of MTS-GFP (Fig 8D) supports the above claim. Observe how diffusive MTS-GFP signal is in the cytoplasm, very different from SepF-GFP. However, it's very exciting to see that MTS-GFP indeed requires FtsZ1 and FtsZ2 to localize. That implies that SepF recognizes the division site in a multifaceted way.

The recommended experiments are very interesting, but difficult to perform as we only have one inducible promoter at hand in *H. volcanii*. As we control the expression of SepF with tryptophan, it is difficult to then induce the expression of the other protein. So, we tried to clone a plasmid where MTS-GFP is under control of the native SepF promoter. Unfortunately, we were not successful to clone the plasmid.

I completely understand the issue. I could recommend alternatives but I believe it would be easier just to tone down the above claims and mention the possible explanations for such intriguing MTS-GFP localization.

Reviewer #2 (Remarks to the Author):

Overall, I am satisfied by the rebuttal of Nussbaum and coworkers to my previous remarks, although the curious targeting of the short membrane targeting sequence of SepF to cell division sites, remains unresolved. The authors show now that FtsZ is required, but that does not exclude my suggestion that it binds to the native SepF, which still seems to be present.

I have to admit that I find the novelty of both the Nussbaum and Pende study limited for [redacted] and also Nat Communications. Both studies confirm the now well established function of the conserved cell division protein SepF.

Reviewer #3 (Remarks to the Author):

I have previously reviewed this paper, and will focus on what the authors did to address my earlier critique. I note that the critique of the other reviewers was extensively addressed.

My critique regarding the presentation of the pulldown experiment (Fig. 8, Fig. S10) has not been addressed – as it cannot be by merely changing the description of the experiment. As stated before: a minimum gel and blot would include a sample of the material applied to the beads, a wash fraction, and an elution fraction, which allows the reader to see how efficient the IP was. This has not been provided, and is considered a must by this reviewer.

The authors describe a series of biochemical experiments that were tried and that were unsuccessful. This is not that surprising – dynamic light scattering does not work for all FtsZs purified, SPR is not very suitable for use with FtsZ that has a great affinity for dextran, which coats most SPR chips, and pull downs require strong interactions. The easiest experiment – sedimentation of FtsZ polymers with or w/o SepF seems to not have been tried. To counteract the high salt higher concentrations of divalent cations (Mg and Ca) could be tried – the gold standard is the nucleotide dependency of the sedimentation. This is not a make or break experiment for this paper in my opinion, but would be worth a try.

The description of the overexpression of SepF and the comparison with other bacteria should be interpreted with more caution (eg line 451-455). The authors show that the promoter used (see also other comment) results in a 1.5 fold overexpression of SepF. I have done a quick check for overexpression levels in the reported studies in Mycobacterium and B. subtilis, but all that I could find was that in both cases the authors mention use of a 'strong' promoter – but this makes it likely that the levels of SepF that result in a block in cell division may be substantially higher than the 1.5 fold in this paper. So it is very positive that the authors have an idea of the extent to which SepF is overproduced, but the direct comparison made with phenotypes observed in other OE experiments in the literature is not completely valid.

L 175 ...not expressed – change to 'not detected'. Also line above – absence of protein does not mean absence of expression per se.

L 221-225 – rephrase.

L242-243 – This should be phrased more cautiously. Suppl. Fig. 4 does not show the overproduction of SepF from pSVA5960, but from a similar plasmid in which SepF-HA is cloned to facilitate quantification of the HA-tag. Thus – levels of untagged are inferred to be the same as levels of SepF-HA expressed from the same promoter. This is likely to be true but in the current version it is stated as proven, which it is not.

L 377 – typo Figure 411B.

REVIEWER COMMENTS

Reviewer #1 (Remarks to the Author):

Old comment from us: The organization of the genomic neighborhood of *hvo_0392* it suggests that these genes are not organized in an operon (see Figure S1). There is a substantial gap of 120 Bp between *hvo_0392* and the downstream ORF (*hvo_0393*). Moreover, transcriptome data which were shared with our group implicate indeed that this is not an operon.

I agree that the appearance is suggestive, but we should be careful with our human common sense as nature is not always ruled by the pattern our eyes are used to see. Transcripts are a great example of how even the most experienced molecular biologists can be wrong without supportive data. I had a lot of fun looking at the literature after datasets, and there are not one but 2 RNAseq datasets that suggest exactly the opposite of the present claims: Laass 2019 (Plos One) and Gelsing 2018 (JBac). Laas and colleagues offer a spreadsheet with detected transcript lengths and *hvo_0392* and *hvo_0393* shown as a single transcript. Gelsing dataset is not so obvious but the dataset can be downloaded from NCBI and the coding transcript length is identical to Laass. It's an interesting observation that a putative DNA repair ATPase is co-transcribed with SepF. Please clarify this in the text.

We thank the reviewer for looking into the RNA seq data sets in such a detail and observing this, as we are not experts in doing so. We have adapted the text, now mentioning the co-transcription of *hvo_0392* and *hvo_0393*.

We have performed the recommended experiment and have added it to the manuscript. It shows that the localization of SepF is dependent on the presence of FtsZ1 (see pictures and Figure 8).

This is not fully supported by Fig 8. Please, note that Δ ftsZ1 and Δ ftsZ1 Δ ftsZ2 cells are deformed, with abnormal aspect ratios. Meanwhile, the presented Δ ftsZ2 cells are elongated rods, maintaining constant width. It is not possible to conclude from this data whether SepF is delocalizing due to morphological defects caused by the lack of FtsZ1 or because SepF is recruited by FtsZ1 itself. In fact, if you observe cells that are not as deformed, SepF is still capable of localization (Fig. 8C).

This strongly suggests that loss of shape control precedes SepF delocalization. Furthermore, the delocalization pattern of MTS-GFP (Fig 8D) supports the above claim. Observe how diffusive MTS-GFP signal is in the cytoplasm, very different from SepF-GFP. However, it's very exciting to see that MTS-GFP indeed requires FtsZ1 and FtsZ2 to localize. That implies that SepF recognizes the division site in a multifaceted way.

We went through the SepF-GFP localization images in the Δ ftsZ1 strain and provided a selection of cells with a "close to" wild-type shape. In these cells SepF-GFP forms clusters, partially also close to the cell center. However, none of these clusters is organized in the defined ring-like structures we observed for SepF-GFP in H26 or the *ftsZ2* deletion strain (see Supplementary Figure 11).

Reviewer #2 (Remarks to the Author):

Overall, I am satisfied by the rebuttal of Nussbaum and coworkers to my previous remarks, although the curious targeting of the short membrane targeting sequence of SepF to cell division sites, remains unresolved. The authors show now that FtsZ is required, but that does not exclude my suggestion that it binds to the native SepF, which still seems to be present.

It is indeed possible that the MTS-GFP construct is still able to interact with the native SepF causing the localization of the site of cell division. To test this, we wanted to express the MTS-GFP in the SepF depletion strain. To control gene expression in *H. volcanii* we are limited to one inducible promoter, the tryptophan inducible ptnA1 promoter that we used to control SepF expression in the SepF depletion strain. To overcome this issue, we wanted to create a plasmid where MTS-GFP is under the control of the native *sepF* promoter. However, so far, we were not able to obtain this plasmid. We will further try this to be able to answer this question in the future.

I have to admit that I find the novelty of both the Nussbaum and Pende study limited for [redacted] and also Nat Communications. Both studies confirm the now well established function of the conserved cell division protein SepF.

Reviewer #3 (Remarks to the Author):

I have previously reviewed this paper, and will focus on what the authors did to address my earlier critique. I note that the critique of the other reviewers was extensively addressed.

My critique regarding the presentation of the pulldown experiment (Fig. 8, Fig. S10) has not been addressed – as it cannot be by merely changing the description of the experiment. As stated before: a minimum gel and blot would include a sample of the material applied to the beads, a wash fraction, and an elution fraction, which allows the reader to see how efficient the IP was. This has not been provided, and is considered a must by this reviewer.

We are sorry that we misunderstood your request on the presentation of the pulldown experiments. We now provide gels and western-blot with the load, flow through, last wash step and elution fraction of cytosol and membrane fraction of either H26 or the SepF-HA expressing strain HTQ236 (see Supplementary Figure 10).

The authors describe a series of biochemical experiments that were tried and that were unsuccessful. This is not that surprising – dynamic light scattering does not work for all FtsZs purified, SPR is not very suitable for use with FtsZ that has a great affinity for dextran, which coats most SPR chips, and pull downs require strong interactions. The easiest experiment – sedimentation of FtsZ polymers with or w/o SepF seems to not have been tried. To counteract the high salt higher concentrations of divalent cations (Mg and Ca) could be tried – the gold standard is the nucleotide dependency of the sedimentation.

This is not a make or break experiment for this paper in my opinion, but would be worth a try.

Thank you very much for this input! We will definitely try the recommend sedimentation assay in the future.

The description of the overexpression of SepF and the comparison with other bacteria should be interpreted with more caution (eg line 451-455). The authors show that the promoter used (see also other comment) results in a 1.5 fold overexpression of SepF. I have done a quick check for overexpression levels in the reported studies in Mycobacterium and *B. subtilis*, but all that I could find was that in both cases the authors mention use of a ‘strong’ promoter – but this makes it likely that the levels of SepF that result in a block in cell division may be substantially higher than the 1.5

fold in this paper. So it is very positive that the authors have an idea of the extent to which SepF is overproduced, but the direct comparison made with phenotypes observed in other OE experiments in the literature is not completely valid.

We adapted the manuscript to indicate that most likely the 1.5 fold higher expression levels in *H. volcanii* is not enough to block cell division and toned down these statements throughout the results section and the discussion.

L 175 ...not expressed – change to ‘not detected’. Also line above – absence of protein does not mean absence of expression per se.

L 221-225 – rephrase.

L242-243 – This should be phrased more cautiously. Suppl. Fig. 4 does not show the overproduction of SepF from pSVA5960, but from a similar plasmid in which SepF-HA is cloned to facilitate quantification of the HA-tag. Thus – levels of untagged are inferred to be the same as levels of SepF-HA expressed from the same promoter. This is likely to be true but in the current version it is stated as proven, which it is not.

L 377 – typo Figure 411B.

We have changed the respective parts in the manuscript.

REVIEWERS' COMMENTS

[The editors assessed the responses to reviewers #1 and #2 and found them satisfactory]

Reviewer #3 (Remarks to the Author):

Thank you for including fig. S10 and modifying some statements.